Report

EMBO
reports

# Atg1 kinase regulates autophagosome–vacuole fusion by controlling SNARE bundling

Saskia Barz[1,2,3,†], Franziska Kriegenburg[1,†], Anna Henning[1], Anuradha Bhattacharya[1,2,3], Hector Mancilla[1], Pablo Sánchez-Martín[1] & Claudine Kraft[1,4,*] (iD)

## Abstract

Autophagy mediates the degradation of cytoplasmic material. Upon autophagy induction, autophagosomes form a sealed membrane around the cargo and fuse with the lytic compartment to release the cargo for degradation. In order to avoid premature fusion of immature autophagosomal membranes with the lytic compartment, this process needs to be tightly regulated. Several factors mediating autophagosome–vacuole fusion have recently been identified. In budding yeast, autophagosome–vacuole fusion requires the R-SNARE Ykt6 on the autophagosome, together with the three Q-SNAREs Vam3, Vam7, and Vti1 on the vacuole. However, how these SNAREs are regulated during the fusion process is poorly understood. In this study, we investigate the regulation of Ykt6. We found that Ykt6 is directly phosphorylated by Atg1 kinase, which keeps this SNARE in an inactive state. Ykt6 phosphorylation prevents SNARE bundling by disrupting its interaction with the vacuolar SNAREs Vam3 and Vti1, thereby preventing premature autophagosome–vacuole fusion. These findings shed new light on the regulation of autophagosome–vacuole fusion and reveal a further step in autophagy controlled by the Atg1 kinase.

**Keywords** Atg1; autophagosome; autophagy; SNARE; Ykt6
**Subject Categories** Autophagy & Cell Death; Post-translational Modifications & Proteolysis

## Introduction

Macroautophagy, hereafter referred to as autophagy, is an intracellular degradation and recycling pathway that is highly conserved among eukaryotes. During autophagy, cellular components are engulfed by a newly formed double-membrane vesicle, the autophagosome. Upon completion, the outer autophagosomal membrane fuses with a lytic compartment, that is, the lysosome in mammals or the vacuole in yeast and plants. Fusion releases the inner vesicle for degradation in the lumen of the lytic compartment.

Both non-selective bulk autophagy of random cytoplasm and selective autophagy of specific cargo have been described. A selective autophagy-related pathway in yeast is the cytoplasm-to-vacuole targeting (Cvt) pathway, which fulfills a biosynthetic function by delivering at least three resident enzymes, aminopeptidase 1 (Ape1), alpha-mannosidase (Ams1), and aspartyl aminopeptidase (Ape4), to the vacuole (Harding *et al*, 1995; Hutchins & Klionsky, 2001; Kraft *et al*, 2009; Yuga *et al*, 2011). Over 40 autophagy-related (Atg) proteins are known to act in these yeast autophagy pathways. Many functional homologues of these players have been identified also in higher eukaryotes (Nakatogawa *et al*, 2009; Yang & Klionsky, 2010). The core Atg proteins are required for most types of autophagy and are known to regulate different steps of the pathway up to autophagosome maturation and fusion with the lytic compartment.

Central and essential for autophagy initiation and conserved from yeast to mammals is the Atg1 kinase complex in yeast and the homologous uncoordinated-51-like kinase (ULK) kinase complex in mammals (Matsuura *et al*, 1997). Atg1/ULK1 belongs to the family of serine–threonine kinases and is part of the core Atg machinery. The presence of Atg1/ULK1 is essential for the correct assembly of the phagophore assembly site (PAS, also called pre-autophagosomal structure), as well as its maturation and subsequent progression of autophagy (Suzuki *et al*, 2007; Hollenstein *et al*, 2019; Eickhorst *et al*, 2020). Therefore, Atg1/ULK1 is viewed as a key regulator of autophagy, acting throughout the pathway. Several Atg1/ULK1 kinase targets have been identified throughout the pathway. Atg9 phosphorylation by Atg1 regulates membrane expansion, and Atg4 phosphorylation by Atg1 controls the maturation of autophagosomes (Papinski *et al*, 2014; Sánchez-Wandelmer *et al*, 2017).

Mature autophagosomes need to become competent for fusion with the lytic compartment, which requires the action of SNARE (soluble N-ethylmaleimide sensitive factor attachment protein receptor) proteins. Generally, SNARE-mediated fusion is achieved by the formation of a bundle of one R-SNARE and three Q-SNARE proteins, promoted by the action of Rab GTPases and tethering complexes.

1  Institute of Biochemistry and Molecular Biology, ZBMZ, Faculty of Medicine, University of Freiburg, Freiburg, Germany
2  Faculty of Biology, University of Freiburg, Freiburg, Germany
3  Spemann Graduate School of Biology and Medicine (SGBM), University of Freiburg, Freiburg, Germany
4  CIBSS—Centre for Integrative Biological Signalling Studies, University of Freiburg, Freiburg, Germany
   *Corresponding author. Tel: +49 0761 203 5221; E-mail: kraft@biochemie.uni-freiburg.de
   †These authors contributed equally to this work

During autophagosome–vacuole fusion in yeast, the R-SNARE Ykt6 acts on the autophagosome and forms a SNARE bundle with the Q-SNAREs Vam3, Vti1, and Vam7 on the vacuole (Bas *et al*, 2018; Gao *et al*, 2018). The role of Ykt6 in autophagy is also conserved in higher eukaryotes (Matsui *et al*, 2018).

Ykt6 has been described to act in multiple membrane trafficking events, such as homotypic vacuolar fusion, retrograde trafficking to the cis-Golgi, and in the carboxypeptidase Y (CPY) pathway (Sogaard *et al*, 1994; McNew *et al*, 1997; Ungermann *et al*, 1999). Ykt6 has also been reported to function early in autophagy as autophagosomes fail to form in Ykt6 mutants, due to a defect in Atg9 vesicle trafficking (Nair *et al*, 2011). Whereas Ykt6 forms a complex with Vam3, Vam7, and Vti1 during autophagosome–vacuole fusion (Bas *et al*, 2018), it has been reported to bundle with other SNARE proteins, Sso1 and Sec9, during early steps of autophagy (Nair *et al*, 2011). These observations suggest that Ykt6 functions in different pathways or during different steps of one pathway, which might be regulated by its ability to bundle with different SNARE proteins.

In general, membrane fusion processes need to be spatiotemporally regulated to avoid premature fusion of immature membranes such as unsealed autophagosomes. How Ykt6 function is regulated during autophagy, and how premature fusion of autophagic membranes with vacuoles is prevented, remains unknown.

The SNARE protein Ykt6 consists of an N-terminal longin domain, a C-terminal SNARE domain, and a C-terminal lipidation site, which is post-translationally farnesylated and can also be reversibly palmitoylated. Furthermore, geranylgeranylation of human Ykt6 has been shown to be important for Golgi integrity and function (Shirakawa *et al*, 2020). Ykt6 is both present in the cytosol and on membranes. Cytosolic Ykt6 adopts a closed and inactive conformation. In this conformation, the N-terminus of Ykt6 is folded back and tightly interacts with the partially folded SNARE domain. The interaction creates an internal hydrophobic core where the farnesyl-moiety is engaged and shielded from the cytosol, keeping Ykt6 soluble. On membranes, Ykt6 is anchored through the insertion of the farnesyl residue into the lipid-bilayer and adopts an open and active conformation, allowing its bundling with other SNAREs and eventually triggering fusion of the associated membranes (Kriegenburg *et al*, 2019). Whether membrane recruitment is sufficient to induce the open state or whether additional regulatory mechanisms exist is not known. Also, how and when Ykt6 reaches its destination membrane, and how Ykt6 is timely regulated to avoid aberrant SNARE assembly, is not understood to date. Likely scenarios for Ykt6 regulation could be either by coordinating its recruitment to membranes, or by controlling its open and closed conformation on the membrane. Here, we show that Atg1 phosphorylation of Ykt6 keeps this SNARE in an inactive state and prevents SNARE bundling by disrupting the interaction with the vacuolar SNAREs Vam3 and Vti1. These findings show that Atg1 kinase regulates also the last step in autophagy, namely autophagosome–vacuole fusion.

## Results and Discussion

### Ykt6 is a direct Atg1 kinase target

Ykt6 is required for both bulk and selective autophagy (Nair *et al*, 2011; Bas *et al*, 2018). Besides Ykt6, none of the other metazoan SNAREs that mediate autophagosomal fusion have homologues in yeast. The high degree of functional conservation of Ykt6 between yeast and mammals points to a central role of this SNARE protein in autophagosome–lytic compartment fusion (Fig 1A). Since the fusion process needs to be timely controlled to avoid premature fusion of unclosed autophagosomes, we hypothesized that Ykt6 function could be regulated by phosphorylation. Sequence analysis of yeast Ykt6 revealed two potential Atg1 consensus sites at Ser182 and Ser183, one being conserved in mammalian YKT6 (Fig 1A, Papinski *et al*, 2014). A recent phosphoproteome study identified these serine residues, along with Thr158, to be phosphorylated in an Atg1-dependent manner *in vivo* (Hu *et al*, 2019). We therefore speculated that Ykt6 function in autophagy could be controlled by Atg1.

To determine whether Atg1 directly phosphorylates Ykt6, we purified Ykt6 from *E. coli*. GST-fusion proteins of Ykt6 lacking the C-terminal acylation sites (Ykt6Δac) and the Ykt6 SNARE domain lacking the acylation sites (SNAREΔac) were generated. As Ser182 and Ser183, but not Thr158 match the Atg1 kinase consensus sequence, we analyzed these phosphorylation sites also individually, and created short amino acid peptides spanning either Ser182 and Ser183 (SS peptide) or Thr158 (T peptide, Fig 1B). The isolated GST-fusion constructs were subjected to *in vitro* phosphorylation using the native Atg1 kinase complex purified from yeast, as described previously (Papinski *et al*, 2014; Pfaffenwimmer *et al*, 2014). All fusion proteins except the T peptide were phosphorylated *in vitro* by Atg1 (Fig 1C). Alanine mutation of these potential target sites in the SNARE domain or in the peptides largely abolished *in vitro* phosphorylation by Atg1, whereas some phosphorylation remained in the Ykt6Δac protein. Together, these findings show that Ser182 and Ser183 in Ykt6, but not Thr158, are directly phosphorylated by Atg1 kinase.

### Ykt6 phosphorylation inhibits autophagy flux

As Ykt6 is a multifunctional SNARE protein important for different intracellular membrane trafficking events, it is not surprising that loss of Ykt6 is lethal in eukaryotes (McNew *et al*, 1997). To study Ykt6 function *in vivo*, we first aimed at generating a functionally tagged Ykt6 allele expressed at native levels, as overexpression of Ykt6 has been reported to result in aberrant cytosolic localization (Meiringer *et al*, 2008). We generated tagged alleles of Ykt6 by fusing GFP or 3HA to its N-terminus. These fusion proteins were expressed under the native Ykt6 promoter to avoid an aberrant overexpression of the protein, which could alter its native membrane association and function. As Ykt6 is essential for viability, fusion proteins were expressed in heterozygous diploid *ykt6Δ/YKT6* knockout cells, and cell viability was tested after dissection of the haploid spores. Whereas the 3HA-Ykt6 fusion protein resulted in four viable spores, GFP-Ykt6 dissections resulted in the growth of only two spores, indicating that the GFP fusion protein is unable to rescue a *YKT6* deletion (Appendix Fig S1A).

Next, we tested the function of these fusion proteins in bulk autophagy by performing a Pho8Δ60 assay, which monitors the autophagic delivery of the cytosolic Pho8Δ60 phosphatase into the vacuole (Noda *et al*, 1995). Cells containing a temperature-sensitive (ts) variant of Ykt6 show a severe autophagy defect at restrictive temperature (Fig 2A, Bas *et al*, 2018). Whereas GFP-Ykt6 failed to restore bulk autophagy in these cells, 3HA-Ykt6 restored autophagy

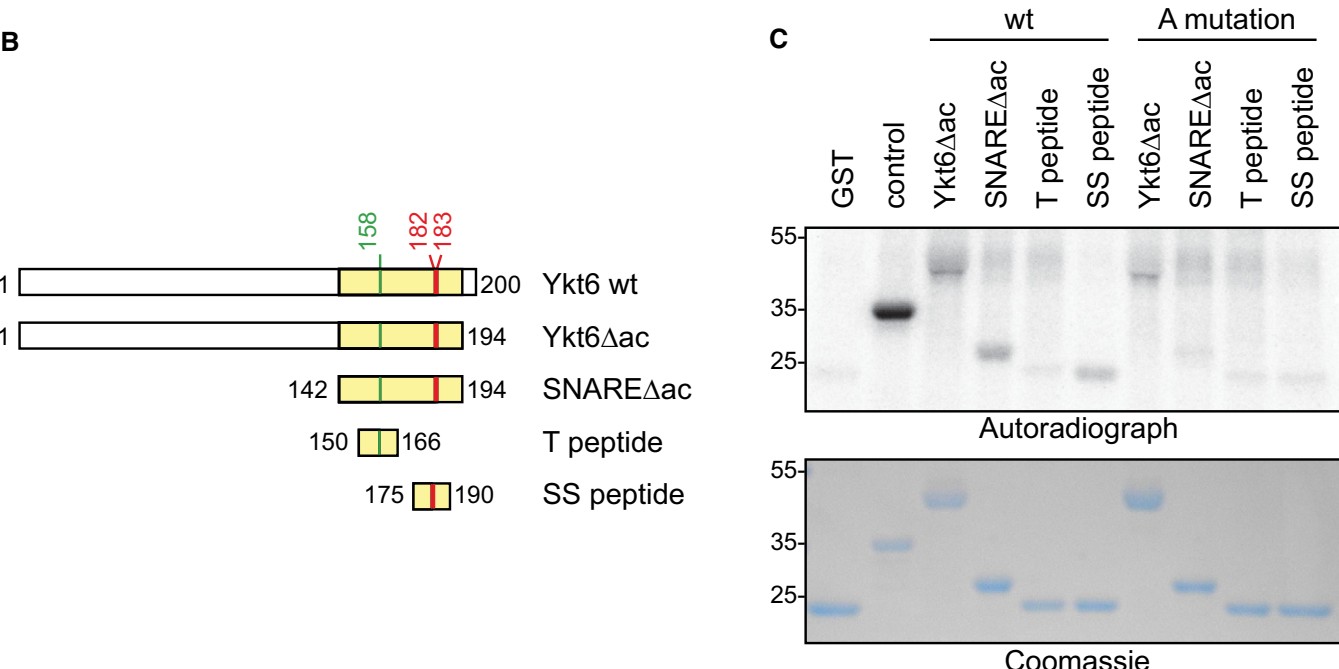

**Figure 1. Ykt6 is a direct Atg1 kinase target.**

A   Alignment of yeast and human Ykt6 protein sequence. The SNARE domain is labeled in yellow, and phosphorylation sites are marked in green and red. Red phosphorylation sites match the Atg1 kinase consensus.

B   Constructs used in *in vitro* kinase assay shown in (C).

C   GST, GST-Atg19 C-terminus (control), GST-Ykt6Δac (1–194 aa), GST-SNAREΔac (142–194 aa), GST-T-peptide, GST-SS-peptide, and alanine (A) mutants were purified from *E. coli* and *in vitro* phosphorylated with immunoprecipitated Atg1-TAP bound to IgG magnetic beads. One representative experiment out of three is shown.

to about 80%, further suggesting that the N-terminal fusion of Ykt6 with 3HA is functional; however, a GFP fusion renders the protein inactive. Therefore, we continued our study with the 3HA-Ykt6 fusion protein.

To investigate whether Atg1-mediated phosphorylation of Ykt6 is important for Ykt6 function *in vivo*, we constructed non-phosphorylatable alanine and phospho-mimicking aspartate mutants. All three phosphorylation sites (Thr158, Ser182, and Ser183) were mutated to alanine (3A) or aspartate (3D) and analyzed for their effect on cell viability. Although Thr158 did not appear to be directly targeted by

Atg1 *in vitro* (Fig 1C), we decided to further analyze this site, as it has been suggested to be phosphorylated in an Atg1-dependent manner *in vivo*. Hence, an aspartate mutation of Thr158 (T158D) and a double aspartate mutant of Ser182 and Ser183 (SSDD) were also analyzed individually. Whereas the 3A mutant was able to complement a *YKT6* deletion, the 3D mutant and the T158D and SSDD mutants were unable to restore cell viability (Appendix Fig S1B and C), suggesting that only mimicking a dephosphorylated state, but not a phosphorylated state can complement Ykt6 function.

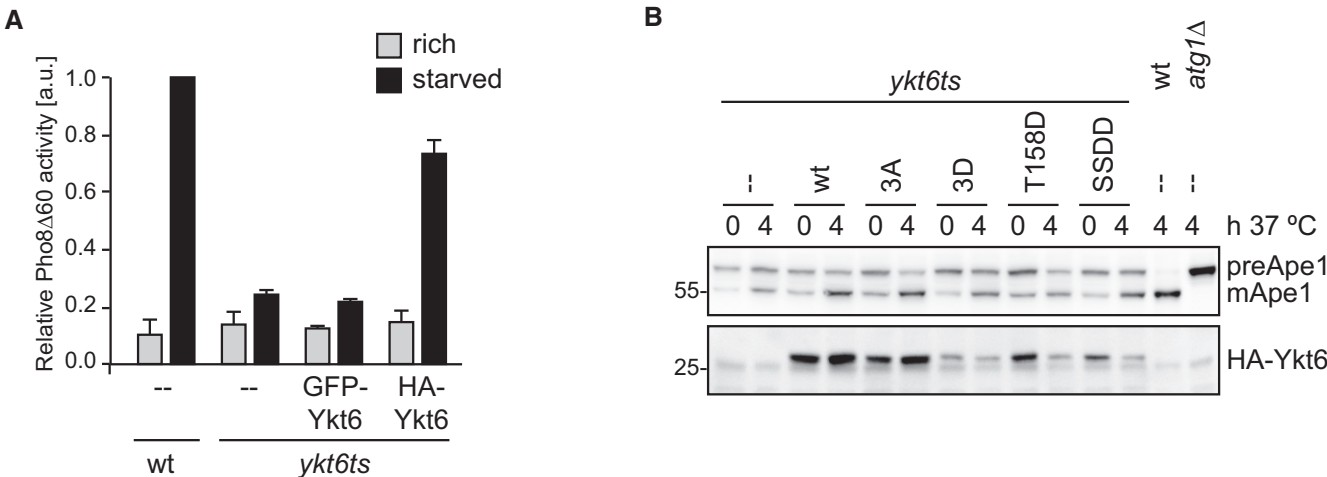

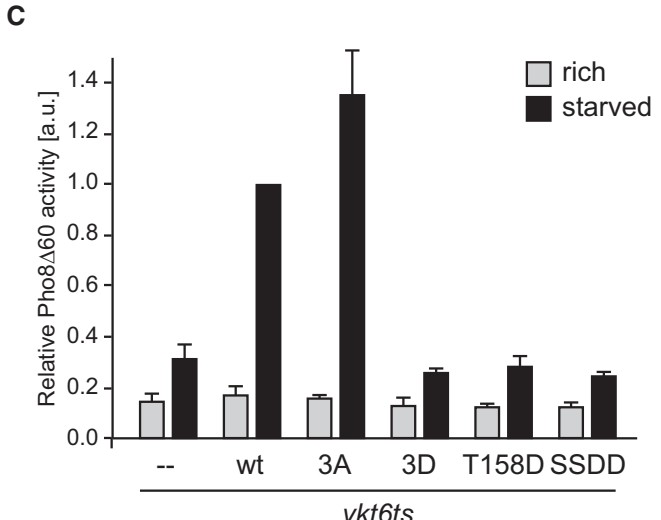

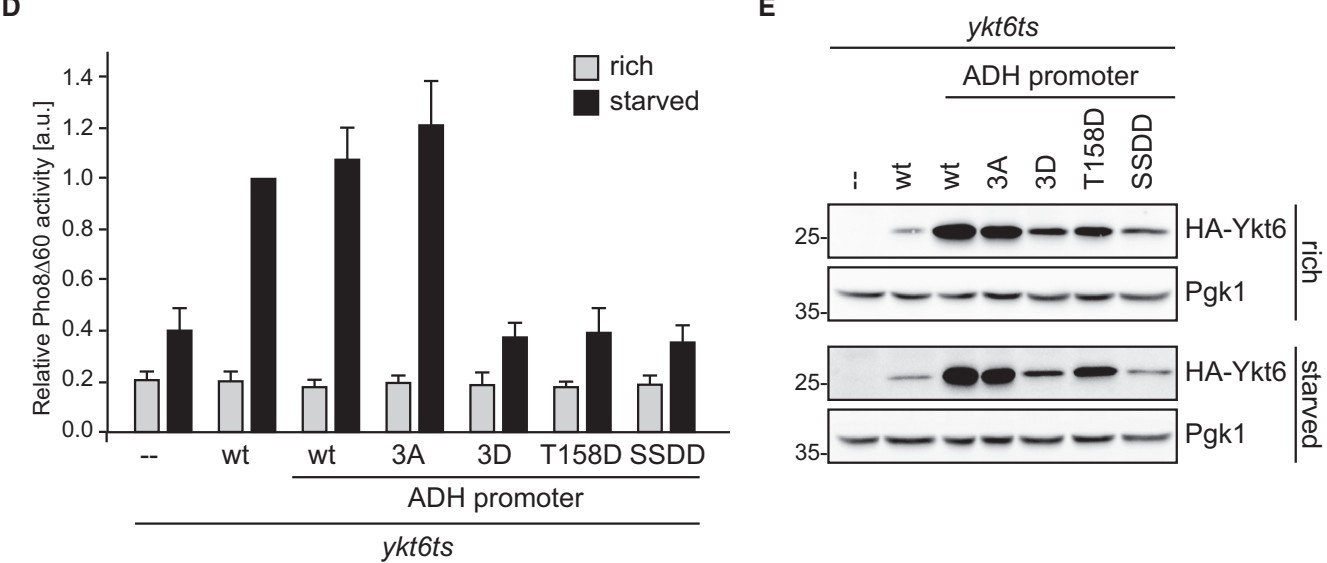

**Figure 2.**

**Figure 2. Ykt6 phosphorylation inhibits autophagy flux.**

A  *ykt6ts pho8Δ60* cells containing an empty plasmid, GFP-Ykt6, or 3HA-Ykt6 on a centromeric plasmid and *pho8Δ60* cells containing an empty plasmid were grown at permissive temperature (23°C) to late exponential phase in selective medium. Cells were shifted to the restrictive temperature (37°C) for 1 h (light gray bars) and subsequently starved for 4 h in SD-N medium at restrictive temperature (black bars). Pho8Δ60 alkaline phosphatase activity was measured in three independent biological experiments, and the mean was plotted normalized to starved Pho8Δ60 alkaline phosphatase activity. Error bars represent the standard deviation.

B  Trichloroacetic acid extracts were prepared from *ykt6ts* cells containing an empty plasmid, 3HA-Ykt6 wild-type, 3HA-tagged Ykt6 mutant variants or *atg1Δ*, and wild-type cells as controls. Cells were grown at the permissive temperature (23°C) to late exponential phase in selective medium. Cells were shifted to the restrictive temperature (37°C) for 1 h preincubation to inactivate the endogenous Ykt6 temperature-sensitive protein, and harvested (0) or further incubated at 37°C for an additional 4 h (4). Samples were analyzed by anti-Ape1 and anti-HA Western blotting. One representative experiment out of three biological replicates is shown. Wild-type or *atg1Δ* cells served as controls. Note that the temperature shift results in enhanced Ape1 processing.

C  *ykt6ts pho8Δ60* cells containing an empty plasmid, 3HA-Ykt6, or 3HA-tagged Ykt6 mutant variants as indicated were analyzed as in (A).

D  *ykt6ts pho8Δ60* cells containing an empty plasmid, 3HA-Ykt6, or ADH1 promoter-driven 3HA-Ykt6 wild-type or 3HA-tagged Ykt6 mutant variants as indicated were grown, and Pho8Δ60 activity was monitored as described in (A).

E  Ykt6 levels were analyzed from the strains shown in (D), and corresponding cell extracts were analyzed by anti-HA and anti-Pgk1 Western blotting.

Next, we analyzed the effect of these mutants on the selective Cvt pathway and bulk autophagy when expressed in *ykt6ts* cells at restrictive temperature. First, we monitored Cvt activity by measuring Ape1 processing. Whereas the 3A mutation had no effect on the Cvt pathway (Fig 2B), *ykt6ts* mutants containing the 3D mutant as well as the T158D and the SSDD variants showed a strong reduction in Ape1 processing, similar to *ykt6ts* cells containing a control plasmid. Next, we investigated the effect of these mutants on bulk autophagy using the Pho8Δ60 assay. *ykt6ts* cells containing the 3A mutant showed even increased autophagic activity compared with *ykt6ts* cells containing wild-type Ykt6, but the 3D mutant was unable to restore autophagy function (Fig 2C). Similarly, the T158D and the SSDD mutants did not restore any significant levels of bulk autophagy. These results were confirmed by a Pgk1-GFP cleavage assay (Welter *et al*, 2010), in which processing of Pgk1-GFP was strongly inhibited by the Ykt6 phospho-mimicking alleles (Appendix Fig S1D). Together, these findings suggest that phosphorylation of Ykt6 inhibits both selective and bulk autophagy progression.

When analyzing the different mutants, we realized that especially upon starvation the protein abundance of the aspartate mutants dropped (Fig 2B). To exclude that the defect in Cvt and bulk autophagy observed is merely due to decreased Ykt6 protein levels, we expressed 3HA-Ykt6 wild-type and the mutants under a stronger *ADH1* promoter. We reassessed the aspartate mutants for their function in bulk autophagy, using the *ADH1*-driven constructs. Similar to the results obtained with Ykt6 mutants expressed under the native promoter, also the *ADH1*-driven expression of the aspartate mutants showed a severe defect in bulk autophagy and the Cvt pathway (Fig 2D and Appendix Fig S1E). Importantly, *ADH1*-driven expression of the aspartate mutants restored native protein levels also under starvation conditions, comparable to wild-type Ykt6 expressed under its native promoter (Fig 2E). Therefore, we conclude that the autophagy defect observed is not due to decreased protein levels but rather due to the phospho-mimicking state of these proteins.

In summary, these results demonstrate that Ykt6 phosphorylation modulates the Cvt pathway and autophagy function *in vivo*.

## Ykt6 phosphorylation does not alter its membrane association

Ykt6 function has been reported to be regulated by a conformational change from a closed cytosolic form to an open membrane-bound form. To test whether Ykt6 membrane association is regulated by phosphorylation, we analyzed *ykt6ts* cells expressing the different Ykt6 mutants in their ability to associate with membranes.

To separate the membrane from the cytosolic fraction, starved yeast cells were lysed and the crude cell extract was separated into a membrane and a cytosolic fraction by centrifugation at 100,000 *g*. We detected the cytosolic marker Pgk1 mainly in the supernatant fractions, whereas the membrane marker Tom70 was enriched in the pellet, indicating that the two fractions were efficiently separated. 3HA-Ykt6 wild-type was detected in the pellet fraction, as expected (Meiringer *et al*, 2008; Fig 3A). None of the Ykt6 mutants altered this localization, suggesting that Ykt6 phosphorylation on these residues does not affect Ykt6 association with membranes. Therefore, Atg1 must regulate Ykt6 function by other means.

## Ykt6 mutants display early autophagy defects

Since Ykt6 membrane association was not altered, we analyzed which autophagy step is affected by the Ykt6 phospho-mimicking mutants. To test whether autophagosomes are formed and sealed, we performed a protease protection assay and monitored the sensitivity of preApe1 to proteinase K treatment by Western blotting. PreApe1 within sealed autophagosomes is protected, whereas preApe1 is processed to a 50 kDa protease-resistant fragment when autophagosome formation is incomplete. We used *vam3Δ* fusion-deficient cells to prevent autophagosome–vacuole fusion and subsequent processing of preApe1 in the vacuole. While preApe1 from *ykt6ts vam3Δ* cells containing Ykt6 wild-type or the 3A mutant was largely protease resistant as expected for mature autophagosomes, preApe1 from the phospho-mimicking mutants showed partial protease sensitivity (Fig 3B). This shows that these mutants contain immature autophagosomes, and therefore suggests a defect in autophagosome formation.

Ykt6 has been described to function early in autophagy, as autophagosomes show a formation defect in *ykt6ts* cells. This is likely due to impaired intracellular trafficking. To test whether the defect of the Ykt6 phospho-mimicking mutants is caused by impairment of other trafficking pathways, we analyzed CPY maturation. CPY is synthesized as a premature form that travels through the ER and Golgi to its final destination in the vacuole, where it matures. Mature CPY and its precursor forms can be monitored as a mobility shift by anti-CPY Western blotting, reflecting CPY transport and maturation. Thus, CPY maturation can be used as a readout for

**A**

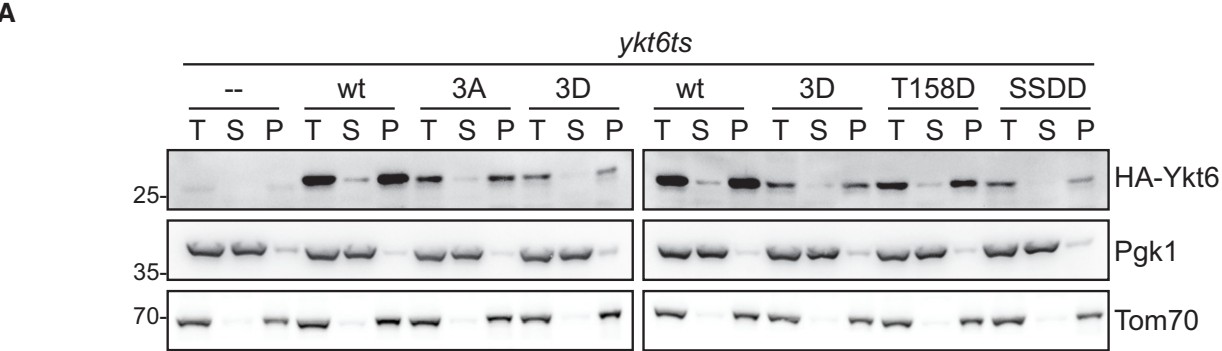

**B**

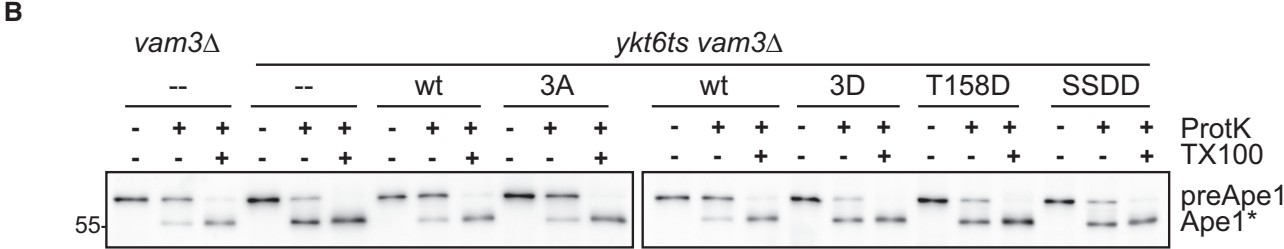

**C**

**D**

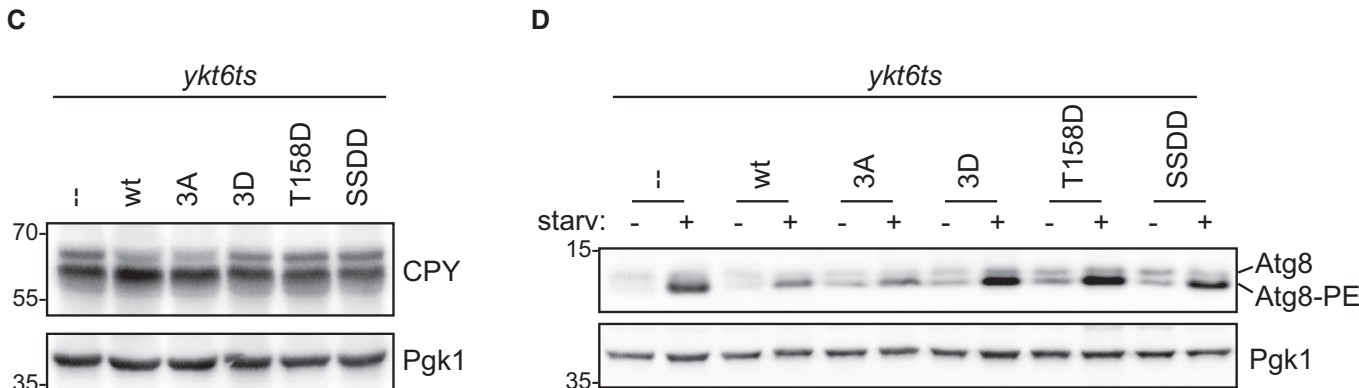

**Figure 3. Ykt6 phosphorylation affects early and late steps in autophagy.**

A  *ykt6ts* cells containing centromeric plasmids as indicated were grown at permissive temperature (23°C), shifted to the restrictive temperature at 37°C for 1 h before starvation for 1 h at 37°C. Cells were lysed and the total extract (T) separated into a cytoplasmic (S) and a 100,000 g membrane pellet (P) fraction. Fractions were analyzed by anti-HA, anti-Tom70, and anti-Pgk1 Western blotting. One representative experiment out of three biological replicates is shown.

B  *vam3Δ* or *ykt6ts vam3Δ* cells containing the indicated centromeric plasmids were grown at permissive temperature (23°C), shifted to the restrictive temperature at 37°C for 1 h followed by 1-h starvation in SD-N at 37°C. After cell lysis, samples were subjected to proteinase K (ProtK) and Triton X-100 (TX100) treatment as indicated and analyzed by anti-Ape1 Western blotting. Ape1*: proteinase K-resistant fragment of Ape1. One representative experiment out of three biological replicates is shown.

C  Trichloroacetic acid extracts were prepared from *ykt6ts* cells containing an empty plasmid, 3HA-Ykt6 wild-type, or 3HA-tagged Ykt6 mutant variants. Cells were grown at the permissive temperature (23°C) to late exponential phase in selective medium, shifted to the restrictive temperature (37°C) for 1 h preincubation and continued to grow at the restrictive temperature for another 4 h before harvesting. Samples were analyzed by anti-CPY and anti-Pgk1 Western blotting. One representative experiment out of three biological replicates is shown.

D  Exponentially growing *ykt6ts* cells at permissive temperature (23°C) containing centromeric plasmids as indicated were shifted to the restrictive temperature (37°C) for 1 h before starvation for 4 h in SD-N. Trichloroacetic acid extracts were prepared. Lipidation of Atg8 was analyzed on a 15% SDS–PAGE containing 6 M urea by anti-Atg8 and anti-Pgk1 Western blotting. One representative experiment out of three biological replicates is shown.

functional intracellular trafficking. As expected under restrictive temperature, *ykt6ts* cells expressing Ykt6 wild type contain mostly matured CPY and very little precursor. *ykt6ts* cells carrying a control plasmid show a decrease in CPY maturation and an accumulation of precursor CPY (Fig 3C). The 3A mutant was proficient in CPY maturation similar to wild-type Ykt6; however, single or multiple

phospho-mimicking mutations of Ykt6 disrupted CPY maturation, similar to the empty plasmid control. These findings suggest that ER-Golgi trafficking is affected by the phospho-mimicking Ykt6 mutants. As autophagy initiation requires a functional ER-Golgi transport to deliver Atg9 vesicles to the PAS, these Ykt6 mutants likely indirectly affect early autophagy steps due to a delivery failure of Atg9 vesicles.

Another essential step for the formation of autophagosomal membranes is the conjugation of Atg8 to phosphatidylethanolamine (PE). This step is independent of Atg9 function and ER-Golgi transport. Therefore, we asked if Ykt6 phosphorylation affects Atg8 lipidation (Suzuki *et al*, 2001). Atg8-PE can be distinguished from non-conjugated Atg8 on urea containing gels. Both wild-type and all Ykt6 mutant cells were proficient in Atg8 conjugation (Fig 3D), suggesting that lipidation is not regulated by Ykt6. However, *ykt6ts* mutant cells containing a control plasmid or the phospho-mimicking Ykt6 mutants accumulated large amounts of lipidated Atg8. These findings suggest that the turnover of lipidated Atg8, and therefore autophagosomal membranes, is blocked, indicating a maturation and/or autophagosome–vacuole fusion defect.

### Ykt6 mutants are defective in autophagosome–vacuole fusion

To evaluate autophagosome formation and turnover in more detail, we compared the frequency of cells with GFP-Atg8 puncta in Vam3-containing *ykt6ts* cells versus fusion-deficient *ykt6ts vam3Δ* mutants after 1 h of starvation. *vam3Δ* cells are deficient in autophagosome–vacuole fusion and therefore accumulate mature autophagosomes. Thus, GFP-Atg8 puncta represent either the PAS or forming or mature autophagosomes.

As expected, the number of GFP-Atg8 puncta per cell increased from around 2 in Vam3-containing wild-type cells to about 7 in fusion-defective *vam3Δ* cells, indicating that in Vam3-containing cells autophagosomes are turned over by fusion with the vacuole after 1 h of starvation (Fig 4A and B). Whereas the 3A mutant showed a similar increase in GFP-Atg8 puncta per cell, all phospho-mimicking mutants showed little increase, suggesting an autophagosome–vacuole fusion defect. Furthermore, we noticed that *ykt6ts vam3Δ* mutants expressing the Ykt6 phospho-mimicking variants accumulated less GFP-Atg8 puncta compared with wild-type or 3A expressing cells, which confirmed our previous findings that autophagosome formation is also affected in the Ykt6 phospho-mimicking mutants (Fig 3B). Taken together, Ykt6 phosphorylation results in both an autophagosome formation as well as a severe autophagosome–vacuole fusion defect.

To test whether Atg1-dependent phosphorylation of Ykt6 directly inhibits autophagosome–vacuole fusion, we used our recently established *in vitro* fusion assay (Bas *et al*, 2018). This assay recapitulates autophagosome–vacuole fusion *in vitro* using isolated yeast vacuoles and an autophagosome-enriched fraction, and thus prevents indirect autophagosome formation defects caused by impairment of other cellular trafficking pathways. As previously described, autophagosomes were isolated from starved and fusion-deficient GFP-Atg8 *vam3Δ* cells. Vacuoles were isolated from cells lacking the vacuolar lipase Atg15, which results in the stabilization of autophagic bodies within the vacuole and allows their visualization. The vacuolar membrane protein Vph1 was tagged with mCherry to visualize the vacuole. We speculated that if Atg1

controls autophagosome–vacuole fusion by phosphorylating Ykt6, then Ykt6 phospho-mimicking mutants should also prevent autophagosome–vacuole fusion *in vitro*. To assess the fusion efficiency of Ykt6 mutants directly, autophagosomes were isolated from *ykt6ts* cells containing wild-type or mutant Ykt6 as indicated. Autophagosomes were allowed to form at the permissive temperature but the fusion reaction was performed at the restrictive temperature. At permissive temperature, *ykt6ts* mutants are fully functional and generate mature and closed autophagosomes, similar to wild-type cells (Bas *et al*, 2018). The wild-type Ykt6 and the 3A mutant mostly restored autophagosome–vacuole fusion, whereas the phospho-mimicking mutants showed a severe defect, similar to the *ykt6ts* mutant containing a control plasmid (Fig 4C and D). Also, addition of purified Atg1 complexes to the reaction abolished fusion (Appendix Fig S1F). Together, these findings suggest that Ykt6 phosphorylation directly regulates autophagosome–vacuole fusion.

### Ykt6 phosphorylation regulates SNARE bundling

As the fusion defect observed is not due to differences in Ykt6 membrane association (Fig 3A), we asked if SNARE bundling could be affected in the Ykt6 mutants. We therefore tested 3HA-Ykt6 interaction with two of its SNARE partners, Vam3 and Vti1. Wild-type Ykt6 and its mutant variants were expressed in *ykt6ts* cells and starved for 1 h at the restrictive temperature. Cells were lysed, 3HA-Ykt6 was enriched on HA agarose, and co-precipitating proteins were assessed by anti-Vam3 and anti-Vti1 Western blotting. As expected, wild-type Ykt6 was capable of co-precipitating both Vam3 and Vti1 (Fig 4E). Also, the 3A mutant precipitated Vam3 and Vti1 in similar amounts. However, although the aspartate mutants were expressed and purified to higher amounts (using the *ADH1* promoter), the interaction with Vam3 and Vti1 was severely decreased, especially for the Ykt6 variants carrying the SSDD mutation. These results suggest that the phosphorylation state of Ykt6 regulates its ability to interact with the vacuolar SNARE proteins.

In summary, our findings reveal that Atg1 phosphorylation of Ykt6 inhibits premature fusion of autophagosomes with vacuoles by preventing the assembly of Ykt6 into a functional SNARE bundle (Fig 5). Previous reports have identified Ykt6 as the R-SNARE acting on autophagosomes during autophagosome–vacuole fusion. How fusion is timely regulated remained unknown. In this work, we uncovered the underlying mechanism of Ykt6 regulation in this fusion process. We find that Ykt6 is tightly regulated by phosphorylation. Whereas Ykt6 membrane association is not affected, its ability to interact with the vacuolar SNARE proteins Vam3 and Vti1 is impaired by phosphorylation, suggesting that Ykt6 phosphorylation negatively affects SNARE bundling.

Atg1 is the kinase responsible for Ykt6 regulation during autophagosome–vacuole fusion, where it phosphorylates Ser182 and Ser183. Parallel work also identified Ykt6 as a target of the Atg1 kinase, and reported similar findings on the effect on autophagosome–vacuole fusion (Gao *et al*, 2020). Thr158 in Ykt6 is phospho-regulated as well, and it also inhibits SNARE bundling in autophagy when mutated to a phospho-mimicking aspartate. The responsible kinase, however, remains elusive. Either a second kinase cooperates in regulating autophagosome–vacuole fusion together with Atg1, or only Atg1 and Ser182/183 phosphorylation are the regulatory factors in autophagy, whereas Thr158 phosphorylation regulates fusion

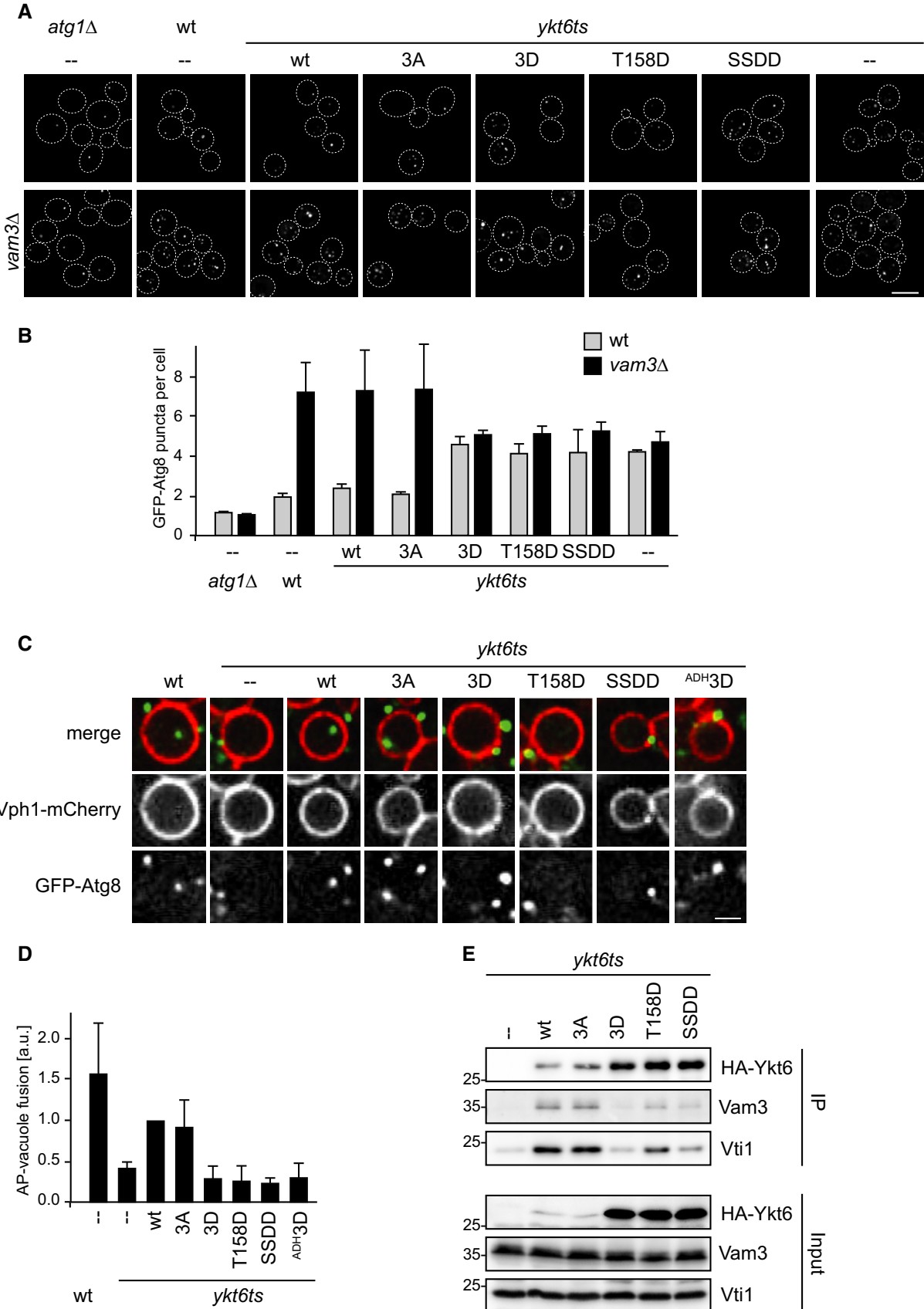

**Figure 4.**

**Figure 4.  Ykt6 phosphorylation prevents autophagosome–vacuole fusion by inhibiting SNARE bundling.**

A, B   GFP-Atg8 *atg1Δ*, GFP-Atg8 *atg1Δ vam3Δ*, GFP-Atg8 wild-type, GFP-Atg8 *vam3Δ*, GFP-Atg8 *ykt6ts,* or GFP-Atg8 *ykt6ts vam3Δ* cells containing centromeric plasmids as indicated were grown at permissive temperature (23°C), shifted to the restrictive temperature at 37°C for 2 h before starvation for 1 h at 37°C in SD-N. Representative microscopy images are shown in (A). GFP-Atg8 puncta per cell were quantified from three biological replicates ($n \geq 100$ cells), and the mean was plotted in (B). Error bars are SD. Scale bar: 5 μm.

C, D   Autophagosomes were prepared from GFP-Atg8 *vam3Δ pep4Δ* or GFP-Atg8 *ykt6ts vam3Δ pep4Δ* cells containing centromeric plasmids as indicated. Cells were starved for 16 h at the permissive temperature (23°C) to allow formation of mature, closed autophagosomes, and shifted for 1 h to the restrictive temperature before harvesting. Vacuoles were isolated from Vph1–4xmCherry *atg15Δ* pep4Δ cells grown under rich conditions at 30°C. Fusion reactions were incubated at the restrictive temperature (30°C) for 2 h. Representative images are shown in (C). Scale bar: 1 μm. The graph in (D) shows the mean from at least three independent biological experiments. Error bars represent the standard deviation.

E      The indicated strains were grown to late exponential phase at permissive temperature (23°C), shifted to the restrictive temperature at 37°C for 1 h followed by starvation in SD-N for 1 h at 37°C. 3HA-Ykt6 and its mutant variants expressed from a centromeric plasmid were immunoprecipitated (IP) and analyzed by anti-HA Western blotting. Co-precipitating proteins were analyzed by anti-Vam3 and anti-Vti1 Western blotting. One representative experiment out of three biological replicates is shown.

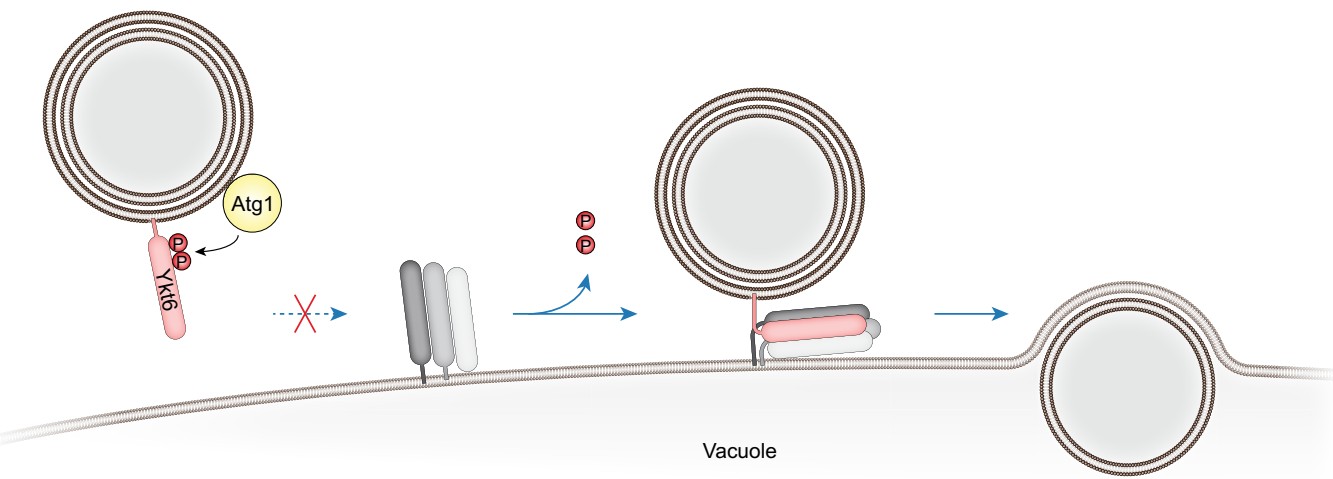

**Figure 5.  Model of Ykt6 phosphoregulation during autophagy.**

Atg1 kinase phosphorylates the SNARE protein Ykt6 on autophagosomes, which prevents interaction of Ykt6 with the vacuolar SNARE proteins. Upon dephosphorylation by an unknown phosphatase, Ykt6 becomes competent for formation of a SNARE bundle with the vacuolar SNAREs Vam3, Vam7, and Vti1, allowing subsequent fusion of the autophagosome with the vacuole.

events in other cellular trafficking pathways. Identification of the kinase responsible for Thr158 phosphorylation will help revealing the underlying mechanism.

It is still unknown how and exactly when Ykt6 is recruited to autophagic membranes. Unphosphorylated Ykt6 might be recruited to forming autophagosomes already. Indeed, parallel work reported the association of Ykt6 to the PAS prior to autophagosome completion (Gao *et al*, 2020). This finding complements and supports our model in which phosphorylated Ykt6 is kept fusion inactive at the nascent autophagosome by preventing SNARE bundle formation until autophagosomes have matured and are ready for the fusion process. Afterward, Ykt6 gets de-phosphorylated by a so far unknown phosphatase, releasing this inhibition and allowing autophagosome–vacuole fusion to take place. This scenario is in analogy to the known phosphoregulation of Atg4 by Atg1, where the Atg4 protease is kept inactive by Atg1 phosphorylation, thereby preventing the premature release of Atg8 from autophagic membranes (Sánchez-Wandelmer *et al*, 2017).

Whereas phosphorylation of Ykt6 prevents its bundling with SNAREs during autophagosome–vacuole fusion, it does not affect the membrane association of Ykt6, suggesting that Ykt6 function is regulated by a structural change on membranes rather than its localization. Whether phosphorylation affects the interaction of Ykt6 with further fusion factors such as the HOPS complex has not been analyzed yet.

## Materials and Methods

### Yeast strains

Yeast strains are listed in Appendix Table S1. Plasmids are listed in Appendix Table S2. Genomic insertions were performed according to Janke *et al*, 2004; Longtine *et al*, 1998; multiple deletions or mutations were generated by PCR knockout or mating and dissection.

## Plasmid construction

For the generation of GFP-Ykt6 (pDP229), 505 bp of the endogenous Ykt6 promoter upstream of the start ATG and the Ykt6 ORF were amplified from genomic yeast DNA. The promoter and ORF were ligated via SacII/XbaI and SbfI/SalI, respectively, into a pRS415 plasmid with an N-terminal GFP cassette and cyc1 terminator (248 bp) present. To obtain the 3HA-Ykt6 construct (pFK33), the GFP tag in pDP299 was replaced by 3HA using overlap extension PCR. Overlapping primers were designed, each coding for part of the 3HA tag, and combined with the forward primer annealing 505 bp upstream in the Ykt6 promoter or at the end of the Ykt6 ORF. The resulting Ykt6pr-3HA-Ykt6 ORF construct was cloned into pDP299 via SacII and SalI and replaced the Ykt6pr-GFP-Ykt6 ORF. The Ykt6 construct was transferred from the pRS415 into pRS416 backbone containing the cyc1 terminator (248 bp) via SacI/XhoI. The cyc1 terminator was finally replaced by the endogenous Ykt6 terminator (300 bp) via SalI and KpnI. The resulting 3HA Ykt6 vector (pFK34) was used as a template to create pRS416 3HA-Ykt6 mutants by mutagenesis PCR. Overexpression constructs of 3HA-Ykt6 variants under the control of the *ADH1* promoter were generated by replacing the endogenous Ykt6 promoter with a 1,500-bp long promoter region upstream of the *ADH1* start ATG via restriction cloning with SacI/SpeI. GST-fusion constructs of Ykt6 were generated by amplifying the region of interest (Ykt6 without acylation site, 1–194 aa, or the SNARE domain without the acylation site) by PCR using either pFK34 or pFK35 as a template and subsequent restriction cloning into pGEX4T1 via BamHI/XhoI. GST-Ykt6 peptides were generated by annealing complementary oligos with sticky BamHI and XhoI ends and ligation into pGEX4T1 via BamHI/XhoI.

## Protein expression

GST and Ykt6 GST-fusion constructs were expressed in BL21(DE3) for 16 h at 16°C. Cells were harvested and washed, and the pellet was resuspended in GST-lysis buffer (PBS; 10% glycerol; 1% Triton X-100 supplemented with cOmplete™ protease inhibitor cocktail (EDTA-free, Roche); and 1 mM PMSF) and lysed via sonication. The cell extract was cleared at 20,000 *g*, 10 min at 4°C, and the resulting supernatant was incubated with GSH Sepharose®4B by end-over-end rotation for 1 h at 4°C. After three washing steps in GST-lysis buffer, the GST-fusion proteins were eluted in elution buffer (50 mM Tris, 10 mM GSH reduced pH 8.0). The isolated proteins were rebuffered in PBS and concentrated using Amicon® Ultra Centrifugal Filter devices.

## Antibodies

The following antibodies were used in this study: polyclonal rabbit anti-Ape1 antibody (1:20,000; Torggler *et al*, 2016), mouse monoclonal anti-Pgk1 antibody (1:20,000; 22C5D8, Invitrogen), rabbit anti-Atg8 antibody (1:500; Bas *et al*, 2018), mouse monoclonal anti-GFP antibody (1:300; 2B6, Merck), rabbit monoclonal anti-HA antibody (1:10,000; EPR4095, Abcam), rabbit anti-Vam3 antibody (1:5000; Scott Emr, Cornell University, USA), rabbit anti-Vti1 antibody (1:5000, Christian Ungermann, University of Osnabrück, Germany), rabbit anti-Tom70 (1:10,000, Chris Meisinger,

University of Freiburg, Germany), and mouse anti-CPY (1:10,000, 10A5, Invitrogen).

## Growth conditions

Yeast cells were grown in synthetic medium (SD: 0.17% yeast nitrogen base, 0.5% ammonium sulfate, 2% glucose, amino acids as required) or rich medium (YPD: 1% yeast extract, 2% peptone, 2% glucose) to mid or late exponential phase as indicated. To induce bulk autophagy, cells were washed and resuspended in nitrogen starvation medium (SD-N: 0.17% yeast nitrogen base without amino acids, 2% glucose). Yeast liquid cultures containing *ykt6ts* cells were grown at the permissive temperature (23°C) or restrictive temperature (30°C or 37°C) as indicated.

## Spot assay

Yeast cell cultures were grown to late exponential phase in selective medium and diluted to an OD600 of 0.4. Fivefold serial dilution series were prepared, and 5 µl of each dilution was spotted onto selective media plates. Pictures were taken after 24 h or 48 h after incubation at 23°C, 30°C, or 37°C.

## Standard biochemical assays

For trichloroacetic acid extract preparation, the indicated OD600 units of yeast cell culture were precipitated with 10% trichloroacetic acid and incubated for 30 min on ice or overnight at −20°C. Precipitated proteins were pelleted at 16,000 *g* for 5 min at 4°C, washed with 1 ml acetone, air-dried, resuspended in 1.5× urea loading buffer (58 mM Tris pH 6.8, 2.45% glycerol, 4 M urea, 71.5 mM β-mercaptoethanol), and boiled, followed by SDS–PAGE. Protein extracts were transferred to nitrocellulose membranes, and proteins were detected by immunoblotting, using the ECL detection system.

## Atg1 kinase assay

Atg1-TAP Atg29-GFP cells were grown to late exponential phase and treated with 220 nM rapamycin for 1 h. Cells were harvested, washed, re-pelleted, and resuspended in 1 ml RLB+/ 1,000 OD600 units (PBS, 10% glycerol, 0.5% Tween 20, 1 mM NaF, 20 mM β-glycerophosphate, 1 mM PMSF, 1 mM Na₃VO₄, cOmplete™ protease inhibitor cocktail (EDTA-free, Roche)). The cell suspension was frozen in liquid nitrogen and subjected to freezer milling. Freezer milled yeast powder was thawed on ice, and 1 ml RLB+/ 1 g powder was added. The extract was pre-cleared by centrifugation (2 times for 10 min, 1,000 g, 4°C), and the resulting supernatant was incubated with IgG-coupled Dynabeads (Thermo Fisher Scientific) for 1 h at 4°C by end-over-end tumbling. The beads were washed three times for 5 min in RLB+ buffer and pre-equilibrated in kinase buffer (20 mM HEPES pH 7.4, 150 mM potassium acetate, 10 mM magnesium acetate, 0.5 mM EGTA, 5 mM NaCl). For the kinase reaction, the isolated Atg1 kinase complex on the Dynabeads was mixed with 3 µg of the various Ykt6 GST-fusion constructs, GST or a GST-tagged Atg19 C-terminal fragment (Pfaffenwimmer *et al*, 2014) and 2 µCi γ-[$^{32}$P]-ATP, and incubated for 30 min at 30°C in kinase buffer supplemented with 10 mM

Na$_3$VO$_4$. The beads and supernatant were separated and mixed with urea loading buffer. Samples were analyzed by SDS–PAGE and phospho-imaging.

## Pho8Δ60 assay

Pho8Δ60 or Pho8Δ60 *ykt6ts* cells were transformed with the indicated plasmids and grown to late exponential phase at 23°C and shifted to 37°C for 1 h, followed by nitrogen starvation at 37°C for 4 h. 20 to 25 OD600 units of yeast culture were harvested by centrifugation (2,000 *g*, 2 min, 4°C) after the 1 h preincubation at 37°C, as well as after nitrogen starvation as indicated. Pellets were washed in 1 ml 0.85% NaCl containing 1 mM PMSF, centrifuged (2,000 *g*, 3 min, 4°C), and snap-frozen in liquid nitrogen. Subsequently, the pellets were resuspended in 16 μl/ OD600 unit lysis buffer (20 mM PIPES pH 6.8, 0.5% Triton X-100, 50 mM KCl, 100 mM potassium acetate, 10 mM MgSO$_4$, 10 μM ZnSO$_4$, 1 mM PMSF, cOmplete™ protease inhibitor cocktail (EDTA-free, Roche)) and cells were lysed by bead beating for 6 min at 4°C. The cell extracts were pre-cleared by centrifugation (16,000 *g*, 5 min, 4°C). Protein concentration of the supernatant was adjusted to 50 μg in 100 μl lysis buffer, and 400 μl reaction buffer (0.4% Triton X-100, 10 mM MgSO$_4$, 10 μM ZnSO$_4$, and 250 mM Tris–HCl pH 8.5) containing 6.25 mM α-naphthyl phosphate (Sigma-Aldrich) was added to enzymatic reactions, or only reaction buffer was added to control reactions. Reactions were incubated at 37°C for 10 min and stopped by adding 500 μl stop buffer (1 M glycine pH 11). Fluorescence was measured using 345 nm for excitation and 472 nm for emission. For each experiment, three independent replicates were performed. Normalized Pho8Δ60 activity was calculated as follows: A standard curve was generated by using a dilution series of the product (1-naphtol, Sigma-Aldrich). Least squares linear regression was performed using the standard curve, which was then used to calculate relative abundance values of the samples. The starvation-induced wild-type sample was set to one, and other samples were normalized accordingly. Mean and standard deviation was calculated with the normalized values.

## In vitro fusion assay

Vacuoles from Vph1-mCherry *atg15Δ pep4Δ* cells were enriched. Cells were grown at 30°C, and a minimum of 1,000 OD600 units were harvested, washed, and treated in DTT containing buffer (100 mM Tris–HCl pH 9.4; 10 mM DTT) for 20 min at 30°C. The cells were spheroplasted in YPD containing 600 mM sorbitol using recombinant lyticase for 30 min at 30°C. Spheroplasts were harvested at 1,500 *g* for 10 min at 4°C, and the pellet was resuspended in 15% Ficoll (in PS200 buffer: 20 mM Pipes, pH 6.8; 200 mM sorbitol supplemented with cOmplete™ protease inhibitor (EDTA-free, Roche)) with 0.08 μg/OD600 unit DEAE-Dextran. The samples were briefly centrifuged at 20,000 *g*, 10 min, 4°C using an ultracentrifuge (Optima MAX-130K Ultracentrifuge (Beckmann)). The pellet was taken up in 15% Ficoll and overlayed with 8%, 4% and 0% Ficoll solution. The gradient was centrifuged at 100,000 *g* for 80 min, 4°C (Sorvall WX Ultracentrifuge). The enriched vacuoles at the 0–4% interface were collected and concentrated at 20,000 *g*, 20 min, 4°C, and finally, the vacuoles were taken up in PS200 buffer.

Autophagosomes were enriched as described before (Bas *et al*, 2018). In brief, GFP-Atg8 *vam3Δ pep4Δ* cells containing an empty vector control and GFP-Atg8 *ykt6ts vam3Δ pep4Δ* strains carrying an empty control plasmid or 3HA-tagged Ykt6 wild-type or its mutant variants were grown at the permissive temperature of 23°C until late exponential phase, the cells were harvested, washed, and starved in SD-N overnight at 23°C. Subsequently, the cells were subjected to 1-h heat treatment at the restrictive temperature of 37°C. The cells were harvested, washed, and treated in DTT containing buffer (100 mM Tris–HCl pH 9.4; 10 mM DTT) for 20 min at 30°C. The cells were spheroplasted in SD-N containing 600 mM sorbitol using recombinant lyticase for 30 min at 30°C. Spheroplasts were harvested at 1,500 *g* for 10 min at 4°C and osmotically lysed in PS200 buffer supplemented with cOmplete™ protease inhibitor (EDTA-free, Roche) and additionally passed through a G20 syringe. The lysate was pre-cleared at 500 *g* for 10 min at 4°C, and the cleared lysate was centrifuged at 20,000 *g*, 20 min, 4°C by ultracentrifugation (Optima MAX-130K Ultracentrifuge). The pellet was taken up in PS200 buffer supplemented with cOmplete™ protease inhibitor (EDTA-free, Roche).

Cytosol was prepared from *ykt6ts* cells as described in Bas *et al* (2018) with small changes. In brief, the cells were grown at the permissive temperature of 23°C until late exponential phase and treated for 1 h with 220 mM rapamycin. Cells were harvested, washed, and re-pelleted. The cell pellet was taken up in 750 μl PS200 buffer/ 1,000 OD600 units, frozen in liquid nitrogen, and subjected to freezer milling. 0.8 g freezer milled yeast powder was thawed on ice, and 40 μl/ 0.1 g PS200 buffer supplemented with cOmplete™ protease inhibitor (EDTA-free, Roche) was added. The extract was pre-cleared by centrifugation (2 times 10 min, 500 *g*, 4°C), and the supernatant was ultracentrifuged at 100,000 *g* for 20 min at 4°C (Optima MAX-130K Ultracentrifuge) and the final supernatant served as cytosol in the fusion assay.

*Ykt6ts* cytosol and enriched crude autophagosomes were treated for 10 min at 37°C before addition to the fusion reactions. Each fusion reaction was set up in 25 μl containing 6 μg enriched vacuoles, 60 μg crude autophagosomes, 60 μg cytosol, and fusion reaction buffer (10 mM Pipes, pH 6.8; 200 mM sorbitol; 125 mM KCl; 5 mM MgCl$_2$; 3 mM ATP; 0,3 mM GTP; cOmplete™ protease inhibitor cocktail (EDTA-free, Roche)). Additionally, an ATP regeneration system (20 mM phosphocreatine (Sigma); 0.5 mg/ml creatine kinase (Sigma)) and 0.15 μg recombinant His-Sec18 and His-Sec17 (gift from Anne Spang, University of Basel, Switzerland) were present. Fusion reactions were incubated at 30°C for 2 h before imaging with a DeltaVision Ultra microscope using time lapse of 45 s with 0.3-s intervals.

## Quantitative live-cell imaging

Fluorescent microscopy images were recorded with a DeltaVision Ultra High Resolution Microscope with UPlanSApo 100×/1.4 oil Olympus objective, using a sCMOS pro.edge camera at room temperature (GE Healthcare, Applied Precision).

To visualize GFP-Atg8 puncta formation, indicated strains were grown to late exponential phase at 23°C in selective media, shifted for 2 h to 37°C before 1-h starvation in SD-N at 37°C. Images were generated by collecting a Z-stack of 15 pictures with focal planes

0.20 μm apart. After deconvolution (SoftWoRx, Applied Precision), image analysis was performed of Z-projections (mean intensity) from ≥100 cells using ImageJ/Fiji software.

Autophagosome–vacuole fusion was followed by time lapse of 45 s with 0.3-s intervals at a signal focal plane. After deconvolution (SoftWoRx, Applied Precision), image analysis was performed and at least 500 vacuoles were counted from minimum three independent experiments using ImageJ/Fiji software.

## Immunoprecipitation

The *ykt6ts* mutant strain was transformed with an empty vector control, 3HA-tagged Ykt6 wild-type, or its mutant variants. Due to reduced expression and immunoprecipitation efficiency, Ykt6 phospho-mimicking variants under the control of the *ADH1* promoter were used. Cells were grown to late exponential phase at 23°C. After 1 h of preincubation at 37°C, cultures were nitrogen-starved for 1 h at 37°C. Approximately 300 OD600 units of yeast culture were harvested by centrifugation (2,000 *g*, 2 min, 4°C) and washed in 5 ml of distilled $H_2O$. Cell pellets were resuspended in 1 ml/ 400 OD600 units RLB+ buffer (PBS, 10% glycerol, 0.5% Tween20, 1 mM NaF, 20 mM β-glycerophosphate, 1 mM PMSF, 1 mM $Na_3VO_4$, cOmplete$^{TM}$ protease inhibitor cocktail (EDTA-free, Roche)) and lysed by bead beating (6 min, 4°C). Lysates were pre-cleared by centrifugation (500 *g*, 5 min, 4°C) and supernatants adjusted to a protein concentration of 2,000 μg in 500 μl RLB+. Input samples were taken, and immunoprecipitation using mouse monoclonal HA agarose (A2095, Sigma Life Science) was conducted with 1,800 μg of protein. After end-over-end tumbling for 1 h at 4°C, the beads were washed 3× in 500 μl RLB+ (end-over-end tumbling, 5 min, 4°C). Finally, beads were taken up in 1.5-fold urea loading buffer (58 mM Tris pH 6.8, 2.45% glycerol, 4 M urea, 71.5 mM β-mercaptoethanol). Samples were subjected to SDS–PAGE and immunoblotting.

## Cell fractionation

The *ykt6ts* mutant strain was transformed with an empty vector control, 3HA-tagged Ykt6 wild-type, or its mutant variants. Cells were grown to late exponential phase at 23°C and shifted to 37°C for 1 h, followed by nitrogen starvation at 37°C for 1 h. After preincubation at 37°C as well as after nitrogen starvation, approximately 6 OD600 units of yeast culture were harvested by centrifugation (2,000 *g*, 2 min, 4°C). Cells were washed in 1 ml ice-cold PBS containing 2% glucose, centrifuged (2,000 *g*, 2 min, 4°C) and nascent pellets resuspended in 200 μl of lysis buffer (PBS, 1 mM NaF, 1 mM NaVO4, 1 mM PMSF, 1 mM EDTA, cOmplete$^{TM}$ protease inhibitor cocktail (EDTA-free, Roche)). Cells were lysed by bead beating (5 min, 4°C), and extracts were pre-cleared by centrifugation (200 *g*, 5 min, 4°C). 100 μl of the cleared cell extracts was used for ultracentrifugation (Optima MAX-130K Ultracentrifuge, 100,000 *g*, 1 h, 4°C).

In preparation for gel electrophoresis, supernatants were mixed with 20 μl 6× SDS sample buffer (375 mM Tris-Cl pH6.8, 12% SDS, 60% glycerol, 0.6% bromophenol blue, 1.8 M β-mercaptoethanol) and pellets resuspended in 120 μl 1× SDS sample buffer. Input samples were prepared by mixing 50 μl of the initial cell extracts with 10 μl of 6× SDS sample buffer. All samples were boiled for 5 min at 95°C, and volumes corresponding to 25 μg of protein of the input sample were subjected to SDS–PAGE and subsequent immunoblotting.

## Protease protection assay

Proteinase K sensitivity of preApe1 was analyzed in GFP-Atg8 *vam3Δ* cells carrying an empty vector control and in GFP-Atg8 *ykt6ts vam3Δ* strains complemented with 3HA-Ykt6 wild-type or the indicated Ykt6 mutant variants next to an empty vector control. The cells were grown in 60 ml selective media to mid/late-log phase (OD600 0.5–1) at 23°C overnight and shifted for 1 h to 37°C. 20 OD600 units of cells were subsequently starved in 25 ml SD-N for 1 h at 37°C. Cells were harvested, washed, treated in 100 mM PIPES pH 9.6, 10 mM DTT for 10 min at 30°C, and finally spheroplasted in SD-N containing 1.2 M sorbitol using recombinant lyticase for 30 min at 30°C. The spheroplasts were pelleted at 500 g at 4°C for 10 min and then subjected to osmotic lysis in 1 ml PS200 lysis buffer (10 mM PIPES pH 6.8, 200 mM sorbitol, 5 mM $MgCl_2$) containing 1 mM PMSF and 1× cOmplete protease inhibitor cocktail (EDTA-free, Roche). The cell extract was pre-cleared three times at 100 g for 5 min at 4°C. The final supernatant was centrifuged at 16,000 *g* for 15 min at 4°C, and the resulting pellet was resuspended in PS200 buffer without protease inhibitors. The sample was aliquoted into three equal parts and incubated as indicated in the presence or absence of proteinase K (100 μg per ml sample) with or without 0.4% Triton X-100 for 20 min on ice. The samples were finally subjected to trichloroacetic acid precipitation and acetone washed.

## Cvt and CPY

Wild-type and *atg1Δ* cells were transformed with an empty vector control. The *ykt6ts* mutant strain was transformed with an empty vector control, 3HA-tagged Ykt6 wild-type, or its mutant variants. Cells were grown to late exponential phase (OD600 0.7–1.5) in 20 ml selective media, shifted for 5 h to the restrictive temperature at 37°C and kept in log phase. 4 OD600 units of cells were harvested after 1 h pre-heat treatment and after additional 4 h of growth at the restrictive temperature. The cells were subjected to trichloroacetic acid precipitation and acetone washed. The processing of Ape1 or CPY was analyzed via SDS–PAGE and Western blotting.

## Pgk1-GFP cleavage and Atg8 lipidation

The *ykt6ts* mutant strain was transformed with an empty vector control, 3HA-tagged Ykt6 wild-type, or its mutant variants, as well as a plasmid for the expression of Pgk1-GFP. Cells were grown to late exponential phase (OD600 0.7–1.5) in 20 ml selective media, shifted for 1 h to the restrictive temperature at 37°C, followed by 4-h starvation in SD-N at 37°C. 4 OD600 units of cells were harvested after the 1 h pre-heat treatment as well as after starvation. The cells were subjected to trichloroacetic acid precipitation and acetone washed. The release of free GFP from Pgk1-GFP during starvation was analyzed via SDS–PAGE and immunoblotting using an anti-GFP serum. Atg8 lipidation was visualized on a 15% SDS–PAGE containing 6 M urea and immunoblotting against Atg8.

# Data availability

This study includes no data deposited in external repositories.

**Expanded View** for this article is available online.

## Acknowledgements

We thank Scott Emr (Cornell University, USA), Christian Ungermann (University of Osnabrück, Germany), and Chris Meisinger (University of Freiburg, Germany) for providing antibodies; Anne Spang (University of Basel, Switzerland) for providing recombinant His-Sec17 and His-Sec18; Ilie Sachelaru for initial experiments; and David Hollenstein for critical reading of the manuscript. We also thank the SGBM graduate school for supporting our students. The Kraft laboratory has received funding from the European Research Council (ERC) under the European Union's Horizon 2020 research and innovation program under grant agreement No 769065, from the European Union's Horizon 2020 research and innovation program under grant agreement No 765912, from the Deutsche Forschungsgemeinschaft (DFG, German Research Foundation) Project-ID 450216812, SFB 1381: Project-ID 403222702, SFB 1177: Project-ID 259130777, and from the EMBO Young Investigator Program. This work reflects only the authors' view and the European Union's Horizon 2020 research, and innovation program is not responsible for any use that may be made of the information it contains. The authors declare no competing financial interests. Open access funding enabled and organized by Projekt DEAL.

## Author contributions

FK, PS, and CK conceptualized the study; SB, FK, and CK contributed to methodology; SB, FK, AH, AB, HM, PS, and CK validated the study; SB, FK, AH, AB, HM, and PS investigated the study, CK wrote—original draft; SB, FK, AH, AB, HM, PS, and CK wrote—review and editing; SB, FK, and CK visualized the study; FK and CK supervised the study; SB, FK, and CK contributed to project administration; CK provided funding acquisition.

## Conflict of interest

The authors declare that they have no conflict of interest.

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
