## [Review Process File · EMBO Reports]

Atg1 kinase regulates autophagosome-vacuole fusion by controlling SNARE bundling

Saskia Barz, Franziska Kriegenburg, Anna Henning, Anuradha Bhattacharya, Hector Mancilla, Pablo Sánchez-Martín, and Claudine Kraft

DOI: [10.15252/embr.202051869](https://doi.org/10.15252/embr.202051869)

Corresponding author(s): Claudine Kraft (claudine.kraft@biochemie.uni-freiburg.de)

Review Timeline:

Submission Date:	11th Oct 20
Editorial Decision:	5th Nov 20
Appeal Received:	6th Nov 20
Editorial Decision:	10th Nov 20
Revision Received:	10th Nov 20
Accepted:	11th Nov 20

Transaction Report:

Dear Claudine,

Thank you once more for the submission of your research manuscript to EMBO reports. We have now received the two reports copied below and I have discussed these further with the referees and the editorial team.

As you will see, referee 1 is overall more positive than referee 2 but also this referee considers further experiments necessary to substantiate the findings and in particular the conclusion that Ykt6 phosphorylation interferes with autophagosome fusion in addition to autophagosome formation. This concern is shared by referee 2 who notes the lack of accumulation of GFP-Atg8 puncta in cells expressing phosphomimetic Ykt6 under certain conditions.

Please note that we have disregarded referee 2's concerns regarding novelty in our decision - as agreed upon - and only considered those concerns that relate to the conclusiveness and technical quality of the study. Given the time-sensitive nature of this submission, we had offered to consider your study under the condition that it can be published with only minor revisions. After reading the reports and upon further discussion with the referees, it became clear that a number of further experiments will be required before publication can be considered. I am therefore sorry to say that we cannot offer to publish your manuscript in EMBO reports.

We have not discussed your work with the editors of our partner journal Life Science Alliance (<http://www.life-science-alliance.org/>; our broad scope Open Access journal published in partnership between the EMBO-, Rockefeller University-, and Cold Spring Harbor Laboratory Presses). I personally feel that your work is an excellent candidate for publication in LSA after it has been revised successfully along the lines suggested by the referees. Please let me know in case you are interested in this option or alternatively, contact the executive editor Shachi Bhatt (s.bhatt@life-science-alliance.org) directly.

From our side, I am very sorry to have to disappoint you and wish you success with the rapid publication of your manuscript.

Kind regards,

Martina

Referee #1:

Barz et al. show that the Atg1 kinase complex phosphorylates Ykt6 in a largely S182/183-dependent manner. The Ykt6 phosphomimic mutants (T158D and S182/183D) do not rescue autophagy and Cvt functions. A protease protection assay suggests that autophagosome formation is affected in Ykt6 T158D and S182/183D cells. To overcome this limitation, the authors

look specifically at the fusion process using their in vitro fusion system and show that fusion is defective for autophagosomes derived from T158D and S182/183D cells. Finally, they show that the T158D and S182/183D mutants interact with Vam3 and Vti1 less efficiently than WT Ykt6.

The manuscript is well-written, and the message is clear. Although similar results were recently published in EMBO Rep by Ungermann's group (Gao et al.), this study provides complementary findings. However, this manuscript contains several issues that need to be addressed.

The authors conclude that the phosphorylation of Ykt6 by Atg1 is important for not only autophagosome formation but also for autophagosome-vacuole fusion based on the results in Fig. 4A (in vivo) and Fig. 4C (in vitro). However, they do not directly show that the autophagic structures in Fig. 4A and Fig. 4C are indeed mature autophagosomes. If they are immature autophagosomes, it would not be surprising that they do not fuse with the vacuole, and would not support the conclusion that autophagosome-vacuole fusion is defective (e.g., in Line 13 on Page 10). This reviewer has some suggestions on how to clarify this, listed as follows:

1. In Fig. 4A, it would be essential to show that these autophagosomes are closed. A protease protection assay should be performed not only for autophagosomes from *vam3d* cells (Fig. 3B) but also from Vam3-containing cells. The result in Fig. 3B that Ape1 is not efficiently protected appears different from that in the paper by Gao et al., in which they show that GFP-Atg8 is mostly protected in *ykt6ts* mutant cells. Electron microscopy may also be helpful.
2. Are all of the autophagosomes in Fig. 4C indeed closed? The protease protection assay would be helpful.
3. Regarding the in vitro fusion assay, it would be best to show that the addition of the Atg1 complex inhibits fusion even when fusion-competent components (not D mutants) are used. This should be feasible as the authors have already purified enzymatic-active Atg1 complex (Fig. 1). A similar experiment was nicely performed by Gao et al. in their paper. It would be the strongest evidence suggesting that Atg1 inhibits premature autophagosome-vacuole fusion.

Finally, it would be fair to state somewhere in the manuscript that Gao et al. also reported that Atg1 phosphorylates Ykt6 to inhibit autophagosome-vacuole fusion.

Referee #2:

Barz et al report that Ykt6 is phosphorylated by Atg1 leading to inactivation of its fusogenic activity.

The sole finding of the manuscript is redundant to the most recently published report by Gao et al. Moreover, while Gao et al further addressed the recruitment of Ykt6 to the autophagosome and its distribution on it, Barz et al only focus on the phosphorylation of Ykt6 by Atg1. While the experiments of Gao et al are nicely designed and for example the effect of Atg1/ Atg13 and ATP on Ykt6 is directly measured in an in vitro assay, the experiments of Barz et al only focus on mutated versions of Ykt6. The lack of GFP-Atg8 accumulation in the Ykt6 phosphomimetic mutants argues against the conclusions drawn. In sum, Gao et al provided significant novel informations about the recruitment and regulation of Ykt6, while the manuscript by Barz et al does not meet the criteria for publication in EMBO Rep. and is solely repetitive.

The manuscript further reads as it was put together in a rush, for example the references are listed in double, but in different versions.

** As a service to authors, EMBO Press provides authors with the ability to transfer a manuscript that one journal cannot offer to publish to another journal, without the author having to upload the manuscript data again. To transfer your manuscript to another EMBO Press journal using this service, please click on

Link Not Available

Point to point reply

As you will see, referee 1 is overall more positive than referee 2 but also this referee considers further experiments necessary to substantiate the findings and in **particular the conclusion that Ykt6 phosphorylation interferes with autophagosome fusion** in addition to autophagosome formation. This concern is shared by referee 2 who notes the lack of accumulation of GFP-Atg8 puncta in cells expressing phosphomimetic Ykt6 under certain conditions.

We can fully address these aspects of fusion, additional experiments have been performed. Please find a detailed response to all points below.

Please note that we have disregarded referee 2's concerns regarding novelty in our decision - as agreed upon - and only considered those concerns that relate to the conclusiveness and technical quality of the study. Given the time-sensitive nature of this submission, we had offered to consider your study under the condition that it can be published with only minor revisions. After reading the reports and upon further discussion with the referees, it became clear that a number of **further experiments will be required** before publication can be considered. I am therefore sorry to say that we cannot offer to publish your manuscript in EMBO reports.

There are no additional experiments needed, we have everything ready to address the raised points by both reviewers.

Referee #1:

Barz et al. show that the Atg1 kinase complex phosphorylates Ykt6 in a largely S182/183-dependent manner. The Ykt6 phosphomimic mutants (T158D and S182/183D) do not rescue autophagy and Cvt functions. A protease protection assay suggests that autophagosome formation is affected in Ykt6 T158D and S182/183D cells. To overcome this limitation, the authors look specifically at the fusion process using their in vitro fusion system and show that fusion is defective for autophagosomes derived from T158D and S182/183D cells. Finally, they show that the T158D and S182/183D mutants interact with Vam3 and Vti1 less efficiently than WT Ykt6.

The manuscript is well-written, and the message is clear. Although similar results were recently published in EMBO Rep by Ungermann's group (Gao et al.), this study provides complementary findings. However, this manuscript contains several issues that need to be addressed.

The authors conclude that the phosphorylation of Ykt6 by Atg1 is important for not only autophagosome formation but also for autophagosome-vacuole fusion based on the results in Fig. 4A (in vivo) and Fig. 4C (in vitro). However, they do not directly show that the autophagic structures in Fig. 4A and Fig. 4C are indeed mature autophagosomes. If they are immature autophagosomes, it would not be surprising that they do not fuse with the vacuole, and would not support the conclusion that autophagosome-vacuole fusion is defective (e.g., in Line 13 on Page 10). This reviewer has some suggestions on how to clarify this, listed as follows:

1. In Fig. 4A, it would be essential to **show that these autophagosomes are closed**. A protease protection assay should be performed not only for autophagosomes from vam3d cells (Fig. 3B) **but also from Vam3-containing cells**.

To address the question of closed autophagosomes, reviewer 1 suggests to test proteinase protection of Ape1 in Vam3 containing cells. In order to test proteinase protection, however, autophagosomes have to be enriched to abolish the turnover of autophagosomes, which otherwise rapidly fuse with the vacuole, hence the use of a vam3delta background. This is a standard procedure done by most labs in the field and has been published before. In contrast, the presence of Vam3 allows autophagosome turnover, which is especially high at 37°C, the temperature needed to induce the ts phenotype of ykt6ts.

Nevertheless, we have done this experiment. In the wild type situation (wild type cells or ykt6ts mutants containing Ykt6 wild type on a plasmid), hardly any preApe1, i.e. autophagosomes, are present, due to their immediate fusion with the vacuole, where preApe1 is also processed. Therefore, little preApe1 can be assessed in the proteinase protection assay in wild type cells. In contrast, ykt6ts mutants containing an empty plasmid do show preApe1, which is protected, supporting the conclusion that these cells contain unfused autophagosomes that are matured. We did not include this in our manuscript since we believe the unideal comparison is more confusing than helpful. This unideal set up is due to the fact that we need to shift the ykt6ts cells to 37 °C, to induce the mutant phenotype. Hence to avoid the high turnover rate at 37 °C and allow comparison, we did the experiment in vam3delta. This setup has the advantage that also early defects are monitored. To address that autophagosome-vacuole fusion is defective, we therefore used other experiments, such as the GFP-Atg8 puncta accumulation and the in vitro assay to clarify the effect of Ykt6 phosphorylation directly in the fusion process.

The result in Fig. 3B that Ape1 is not efficiently protected appears different from that in the paper by Gao et al., in which they show that GFP-Atg8 is mostly protected in *ykt6ts* mutant cells. Electron microscopy may also be helpful.

There are no significant differences in these protection levels when compared to our experiments in the Vam3 containing cells shown here. Similar to Gao et al, a known fusion mutant (Gao et al. used *ypt7delta*, we used *vti1ts*) resulted in similar protection levels as the *ykt6* mutant, see Figure above.

The apparent differences noted by the reviewer are due to the use of fusion deficient *vam3delta* cells in our setup. This setup has the advantage that also early defects are monitored. We clearly explain and write in the manuscript that *ykt6* mutants affect both early autophagosome formation as well as fusion. Therefore, in living cells, these two effects cannot be easily separated. This is the reason why we turned to the in vitro fusion assay, allowing us to directly monitor the involvement of Ykt6 in fusion only. In this fusion assay, fully mature and closed autophagosomes are used, see below.

2. Are all of the autophagosomes in Fig. 4C indeed closed? The protease protection assay would be helpful.

Yes, they are all mature and closed, as they were purified at permissive 23°C, where Ykt6ts is fully functional and behaves as Ykt6 wild type. That Ykt6ts is fully functional at permissive temperature, we and others have shown previously, e.g. Bas et al. JCB 2018, Figure 6B: *ykt6ts* is fully competent in bulk autophagy at 24°C (for which fusion competent and closed autophagosomes have to be formed at normal rates):

[Figure removed]
From Bas et al., 2018, Figure 6B

Also, the Ykt6 A and D mutants do not show a dominant negative effect over the endogenous Ykt6ts protein at permissive temperature, as all mutants are viable and show normal growth rates at permissive temperature. Therefore, purification of autophagosomes at permissive temperature, at which Ykt6ts is fully functional, results in mature and closed autophagosomes.

In short, the autophagosomes are purified at permissive temperature, where in all Ykt6 mutant strains the endogenous Ykt6ts is fully functional and mature and closed autophagosomes are formed. Only during the in vitro reconstitution, the sample is shifted to restrictive temperature to induce the mutant phenotype.

We appreciate that this was not explained enough in the current manuscript and will state this more clearly.

3. Regarding the in vitro fusion assay, it would be best to show that the addition of the Atg1 complex inhibits fusion even when fusion-competent components (not D mutants) are used. This should be feasible as the authors have already purified enzymatic-active Atg1 complex (Fig. 1). A similar experiment was nicely performed by Gao et al. in their paper. It would be the strongest evidence suggesting that Atg1 inhibits premature autophagosome-vacuole fusion.

We have done this experiment (n=3) and can include it right away in our manuscript.

However, Atg1 is known to regulate several proteins in autophagy like Atg4. It is unclear how long Atg4 is kept inactive by Atg1 to avoid premature delipidation of Atg8. It is also unclear if other factors are regulated by Atg1 during fusion. However, we believe that **using the D and A mutants directly in the fusion assay is a more direct way to show where and how Atg1 controls the fusion process: by regulating Ykt6**. Hence, we on purpose concentrated on the phospho-mimicking mutants of Ykt6 to show the specific and direct effect of this modification in fusion.

Finally, it would be fair to state somewhere in the manuscript that Gao et al. also reported that Atg1 phosphorylates Ykt6 to inhibit autophagosome-vacuole fusion.

We agree with this suggestion and will refer to Gao et al. 2020, thereby also highlighting that the two stories strongly support each other, but also complement each other in additional aspects. Whereas Gao et al. have addressed the role of the DSL complex, we add the biological explanation why the phosphorylation of Ykt6 impairs autophagosome-vacuole fusion, namely by inhibiting SNARE bundling. This has never been shown before and is the crucial finding to explain the defects observed, which would otherwise remain elusive.

Referee #2:

Barz et al report that Ykt6 is phosphorylated by Atg1 leading to inactivation of its fusogenic activity. The sole finding of the manuscript is redundant to the most recently published report by Gao et al. Moreover, while Gao et al further addressed the recruitment of Ykt6 to the autophagosome and its distribution on it, Barz et al only focus on the phosphorylation of Ykt6 by Atg1.

We disagree with this statement. The two stories strongly support each other, but also complement each other in additional aspects. Whereas Gao et al. have addressed the role of the DSL complex, we add the **biological explanation why the phosphorylation of Ykt6 impairs autophagosome-vacuole fusion, namely by inhibiting SNARE bundling**. This has never been shown before and is the crucial finding to explain the defects observed, which would otherwise remain elusive. Also, in contrast to Gao et al. **we used functionally tagged Ykt6 expressed at endogenous levels**, whereas Gao et al. overexpressed GFP-tagged Ykt6, which is unable to complement the mutant when expressed at normal levels. Therefore, we feel that both manuscripts support each other but also bring in individual important novel aspects.

While the experiments of Gao et al are nicely designed and for example the effect of Atg1/ Atg13 and ATP on Ykt6 is directly measured in an in vitro assay, the experiments of Barz et al only focus on mutated versions of Ykt6.

We have done this experiment (n=3) and can include it right away in our manuscript.

However, Atg1 is known to regulate several proteins in autophagy like Atg4. It is unclear how long Atg4 is kept inactive by Atg1 to avoid premature delipidation of Atg8. It is also unclear if other factors are regulated by Atg1 during fusion. However, we believe that **using the D and A mutants directly in the fusion assay is a more direct way to show where and how Atg1 controls the fusion process: by regulating Ykt6**. Hence, we on purpose concentrated on the phospho-mimicking mutants of Ykt6 to show the specific and direct effect of this

modification in fusion. The ATP dependence has been tested and reported earlier (Bas et al., JCB 2018; Hollenstein et al. 2019) and has been included here.

The lack of GFP-Atg8 accumulation in the Ykt6 phosphomimetic mutants argues against the conclusions drawn.

Ykt6 mutants have both a formation and a fusion defect. Therefore, in a protease protection assay in vam3D mutants one sees the autophagosome formation defect, as expected when autophagosome formation is slowed down.

This is also nicely reflected in the in vivo assay looking at GFP-Atg8 dot formation in wild type and vam3D fusion deficient cells: Whereas more dots are visible for the Ykt6 mutant in fusion competent cells, which indicates a fusion defect, the number of GFP dots doesn't increase as much as in a wild type cell when fusion is blocked by vam3D, suggesting that also formation of autophagosomes is reduced. This is in agreement with all our other experiments. This is the reason why the role in fusion cannot be studied in vivo, and we turned to the in vitro setup to clearly address the fusion step only. As we purify autophagosomes at permissive temperature in the in vitro setup, these autophagosomes are fully matured. We and others have shown previously, that ykt6ts mutants are fully functional in autophagy at permissive temperature (Bas et al. JCB 2018, Figure 6B).

[Figure removed]
From Bas et al., 2018, Figure 6B

Therefore, what we see here is absolutely expected and in agreement with all other data. We believe that Ykt6 plays a role both in autophagosome formation and during fusion. In this manuscript, however, we focus mainly on the role of Ykt6 in fusion. We appreciate that this needs to be stated more clearly in the text.

In sum, Gao et al provided significant novel informations about the recruitment and regulation of Ykt6, while the manuscript by Barz et al does not meet the criteria for publication in EMBO Rep. and is solely repetitive.

We are aware of the partial overlap, as these studies were performed in parallel without knowing of each other. We disagree with the statement on lacking unique aspects. The two stories strongly support each other, but also complement each other in additional aspects. Whereas Gao et al. have addressed the role of the DSL complex, we add the **biological explanation why the phosphorylation of Ykt6 impairs autophagosome-vacuole fusion, namely by inhibiting SNARE bundling**. This has never been shown before and is the crucial finding to explain the defects observed, which would otherwise remain elusive. Also, in contrast to Gao et al. **we used functionally tagged Ykt6 expressed at endogenous levels**, whereas Gao et al. overexpressed GFP-tagged Ykt6, which is unable to complement the mutant. Therefore, we feel that both manuscripts support each other but also bring in individual important novel aspects.

The manuscript further reads as it was put together in a rush, for example the references are listed in double, but in different versions.

As appreciated by Reviewer 1, this manuscript was very carefully written and also all experiments have been well controlled. We are sorry that the references are listed in double, which must have happened during the final editing with the citation program.

Dear Claudine,

Thank you for your letter asking us to reconsider our decision and invite revision of your manuscript. We handled your manuscript under our scooping protection policy and offered to consider it in case publication was possible with minor revisions only. Referee 1 had suggested a number of important control experiments and you have now informed us that you have already performed these experiments. You have also provided a point-by-point response to the referee's concerns. I have meanwhile discussed your response with the editorial team and with referee 1. Referee 1 evaluated your rebuttal and the new data included and felt that you have adequately addressed all concerns and that your study can be published if the new data are included in the revised version to substantiate your conclusions. Please find the comments from referee 1 pasted below my signature.

As I informed you yesterday already, we therefore decided to invite you to submit a revised manuscript for publication in EMBO reports. In order to publish your manuscript in our December issue, we will need the revised files as soon as possible. Please let me know whether this is possible.

Please upload your point-by-point response and address all concerns from the referees in your revised manuscript. The new data needs to be included. Please also add a reference to Gao et al and a discussion of their findings.

From the editorial side there are also a number of things that we need:

- Please submit a .docx formatted version of the manuscript text and
- production quality figure files as .eps, .tif, .jpg (one file per figure).
- You have currently five figures and your manuscript will be published as Report. This requires that you combine the Results and Discussion section.
- Please add a section on Author Contributions.
- Please add a Conflict of Interest section.
- Please add a Data availability section at the end of Materials and Methods. You can state that you have no data that needs to be deposited in a public repository. (See also <https://www.embopress.org/page/journal/14693178/authorguide#dataavailability>). Please note that the Data Availability Section is restricted to new primary data that are part of this study.
- Please reduce the number of keywords to five.
- Please reformat the references to 10 authors et al
- Supplementary information: You can keep the information as a single .pdf but please change the nomenclature to "Appendix" and "Appendix Figure S1", "Appendix Table S1". You also need a title page with a table of content (including page numbers).
- Alternatively, you can promote Figure S1 to the Expanded View format. In this case we need

individual figure files (Figure EV1 etc) and their legends need to be part of the main manuscript text in a section after the main figure legends. See also

<<https://www.embopress.org/page/journal/14693178/authorguide#expandedview>>

- Please provide a complete author checklist, which you can download from our author guidelines (<<https://www.embopress.org/page/journal/14693178/authorguide>>). Please insert information in the checklist that is also reflected in the manuscript. The completed author checklist will also be part of the RPF.

- Please specify in all figure legends:

- Quantifications: the number of independent experiments (biological or technical replicates), the nature of the bars (mean, median) and the error bars.

- All microscopy images need scale bars. The size needs to be determined in the legend.

- Figure 1C: please define SNARE-delta-ac in the legend.

- Figure 2C, D: please define the number of independent experiments, the nature of the bars and error bars. In case the same applies to 2A,C, D you can also supply this information (n, bars, error bars) in a paragraph called 'Data information: ..' at the end of this legend.

- Figure 4A, C need scale bars

- Figure 4B: How many experiments is this based on? (biological, technical replicate)

- You need to specify all funding in our online submission system too. The following funds have not been entered:

"The Kraft laboratory has received funding from the European Research Council (ERC) under the European Union's Horizon 2020 research and innovation programme under grant agreement No 769065, from the European Union's Horizon 2020 research and innovation programme under grant agreement No 765912, from the Deutsche Forschungsgemeinschaft (DFG, German Research Foundation) Project-ID 450216812, SFB 1381: Project-ID 403222702, SFB 1177: Project-ID 259130777, and from the EMBO Young Investigator Program."

- Finally, EMBO reports papers are accompanied online by A) a short (1-2 sentences) summary of the findings and their significance, B) 2-3 bullet points highlighting key results and C) a synopsis image that is 550x200-600 pixels large (width x height) in .png format. You can either show a model or key data in the synopsis image. Please note that the size is rather small and that text needs to be readable at the final size. Please send us this information along with the revised manuscript.

- As part of the EMBO publication's Transparent Editorial Process, EMBO reports publishes online a Review Process File to accompany accepted manuscripts. This File will be published in conjunction with your paper and will include the referee reports, your point-by-point response and all pertinent correspondence relating to the manuscript.

I look forward to seeing a revised version of your manuscript when it is ready. Please let me know if you have questions or comments regarding the revision.

Kind regards,

Martina

Referee 1

I think these responses would be sufficient. I highly encourage them to include these new data, which are quite important to support the authors' conclusion.

Point-by-point reply

Referee #1:

Barz et al. show that the Atg1 kinase complex phosphorylates Ykt6 in a largely S182/183-dependent manner. The Ykt6 phosphomimic mutants (T158D and S182/183D) do not rescue autophagy and Cvt functions. A protease protection assay suggests that autophagosome formation is affected in Ykt6 T158D and S182/183D cells. To overcome this limitation, the authors look specifically at the fusion process using their *in vitro* fusion system and show that fusion is defective for autophagosomes derived from T158D and S182/183D cells. Finally, they show that the T158D and S182/183D mutants interact with Vam3 and Vti1 less efficiently than WT Ykt6.

The manuscript is well-written, and the message is clear. Although similar results were recently published in EMBO Rep by Ungerermann's group (Gao et al.), this study provides complementary findings. However, this manuscript contains several issues that need to be addressed.

The authors conclude that the phosphorylation of Ykt6 by Atg1 is important for not only autophagosome formation but also for autophagosome-vacuole fusion based on the results in Fig. 4A (*in vivo*) and Fig. 4C (*in vitro*). However, they do not directly show that the autophagic structures in Fig. 4A and Fig. 4C are indeed mature autophagosomes. If they are immature autophagosomes, it would not be surprising that they do not fuse with the vacuole, and would not support the conclusion that autophagosome-vacuole fusion is defective (e.g., in Line 13 on Page 10). This reviewer has some suggestions on how to clarify this, listed as follows:

1. In Fig. 4A, it would be essential to **show that these autophagosomes are closed**. A protease protection assay should be performed not only for autophagosomes from *vam3Δ* cells (Fig. 3B) **but also from Vam3-containing cells**.

To address the question of closed autophagosomes, reviewer 1 suggests to test protease protection of Ape1 in Vam3-containing cells. In order to test protease protection, however, autophagosomes have to be enriched to abolish the turnover of autophagosomes, which otherwise rapidly fuse with the vacuole, hence the use of a *vam3Δ* background. This is a standard procedure done by most labs in the field and has been published before. In contrast, the presence of Vam3 allows autophagosome turnover, which is especially high at 37 °C, the temperature needed to induce the *ts* phenotype of *ykt6ts*.

Nevertheless, we have done this experiment. In the wild type situation (wild type cells or *ykt6ts* mutants containing Ykt6 wild type on a plasmid), hardly any preApe1, i.e. autophagosomes, are present, due to their immediate fusion with the vacuole, where preApe1 is also processed. Therefore, little preApe1 can be assessed in the protease protection assay in wild type cells. In contrast, *ykt6ts* mutants containing an empty plasmid do show preApe1, which is protected, supporting the conclusion that these cells contain unfused autophagosomes that are matured. We did not include this in our manuscript since we believe the unideal comparison is more confusing than helpful. This unideal setup is due to the fact that we need to shift the *ykt6ts* cells to 37 °C, to induce the mutant phenotype. Hence to avoid the high turnover rate at 37 °C and allow comparison, we did the experiment in *vam3Δ*. This setup has the advantage that also early defects are monitored. To address that autophagosome-vacuole fusion is defective, we therefore used other experiments, such as the GFP-Atg8 puncta accumulation and the *in vitro* fusion assay to clarify the effect of Ykt6 phosphorylation directly in the fusion process.

The result in Fig. 3B that Ape1 is not efficiently protected appears different from that in the paper by Gao et al., in which they show that GFP-Atg8 is mostly protected in *ykt6ts* mutant cells. Electron microscopy may also be helpful.

There are no significant differences in these protection levels when compared to our experiments in the Vam3-containing cells shown here. Similar to Gao et al, a known fusion mutant (Gao et al. used *ypt7Δ*, we used *vti1ts*) resulted in similar protection levels as the *ykt6* mutant, see Figure above.

The apparent differences noted by the reviewer are due to the use of fusion deficient *vam3D* cells in our setup. This setup has the advantage that also early defects are monitored. We clearly explain and write in the manuscript that *ykt6* mutants affect both early autophagosome formation as well as fusion. Therefore, in living cells, these two effects cannot be easily separated. This is the reason why we turned to the *in vitro* fusion assay, allowing us to directly monitor the involvement of Ykt6 in fusion only. In this fusion assay, fully mature and closed autophagosomes are used, see below.

2. Are all of the autophagosomes in Fig. 4C indeed closed? The protease protection assay would be helpful.

Yes, they are all mature and closed, as they were purified at permissive 23 °C, where Ykt6ts is fully functional and behaves as Ykt6 wild type. That Ykt6ts is fully functional at permissive temperature, we and others have shown previously, e.g. Bas et al. JCB 2018, Figure 6B: *ykt6ts* is fully competent in bulk autophagy at 24 °C (for which fusion competent and closed autophagosomes have to be formed at normal rates):

Also, the Ykt6 A and D mutants do not show a dominant negative effect over the endogenous Ykt6ts protein at permissive temperature, as all mutants are viable and show normal growth rates at permissive temperature. Therefore, purification of autophagosomes at permissive temperature, at which Ykt6ts is fully functional, results in mature and closed autophagosomes.

In short, the autophagosomes are purified at permissive temperature, where in all Ykt6 mutant strains the endogenous Ykt6ts is fully functional and mature and closed autophagosomes are formed. Only during the *in vitro* reconstitution, the sample is shifted to restrictive temperature to induce the mutant phenotype.

We appreciate that this was not explained enough in the current manuscript and have stated this more clearly in the revised manuscript:

"At permissive temperature, ykt6ts mutants are fully functional and generate mature and closed autophagosomes, similar to wild type cells (Bas et al. 2018)."

3. Regarding the *in vitro* fusion assay, it would be best to show that the addition of the Atg1 complex inhibits fusion even when fusion-competent components (not D mutants) are used. This should be feasible as the authors have already purified enzymatic-active Atg1 complex (Fig. 1). A similar experiment was nicely performed by Gao et al. in their paper. It would be the strongest evidence suggesting that Atg1 inhibits premature autophagosome-vacuole fusion.

We have done this experiment (n=3) and have included it in the new Appendix Figure S1F. We also refer to this experiment in the revised text:

"Also, addition of purified Atg1 complexes to the reaction abolished fusion (Appendix Figure S1F)."

However, Atg1 is known to regulate several proteins in autophagy like Atg4. It is unclear how long Atg4 is kept inactive by Atg1 to avoid premature delipidation of Atg8. It is also unclear if other factors are regulated by Atg1 during fusion. Thus, we believe that **using the D and A mutants directly in the fusion assay is a more direct way to show where and how Atg1 controls the fusion process: by regulating Ykt6**. Hence, we on purpose concentrated on the phospho-mimicking mutants of Ykt6 to show the specific and direct effect of this modification in fusion.

Finally, it would be fair to state somewhere in the manuscript that Gao et al. also reported that Atg1 phosphorylates Ykt6 to inhibit autophagosome-vacuole fusion.

We agree with this suggestion and have changed the manuscript accordingly. We refer to Gao et al. 2020, and also highlighting that the two stories strongly support each other, but also complement each other in additional aspects. Whereas Gao et al. have addressed the role of the DSL complex, we add the biological explanation why the phosphorylation of Ykt6 impairs autophagosome-vacuole fusion, namely by inhibiting SNARE bundling. This

has never been shown before and is the crucial finding to explain the defects observed, which would otherwise remain elusive.

“Recent parallel work also identified Ykt6 as a target of the Atg1 kinase, and reported similar findings on the effect on autophagosome-vacuole fusion (Gao et al. 2020).”

and

“Indeed, recent work reported the association of Ykt6 to the PAS prior to autophagosome completion (Gao et al. 2020). The finding complements and supports our model in which phosphorylated Ykt6 is kept fusion inactive at the nascent autophagosome by preventing SNARE bundle formation until autophagosomes have matured and are ready for the fusion process.”

Referee #2:

Barz et al report that Ykt6 is phosphorylated by Atg1 leading to inactivation of its fusogenic activity. The sole finding of the manuscript is redundant to the most recently published report by Gao et al. Moreover, while Gao et al further addressed the recruitment of Ykt6 to the autophagosome and its distribution on it, Barz et al only focus on the phosphorylation of Ykt6 by Atg1.

We disagree with this statement. The two stories strongly support each other, but also complement each other in unique aspects. Whereas Gao et al. have addressed the role of the DSL complex, we add the **biological explanation why the phosphorylation of Ykt6 impairs autophagosome-vacuole fusion, namely by inhibiting SNARE bundling**. This has never been shown before and is the crucial finding to explain the defects observed, which would otherwise remain elusive. Also, in contrast to Gao et al. **we used functionally tagged Ykt6 expressed at endogenous levels**, whereas Gao et al. overexpressed GFP-tagged Ykt6. GFP-Ykt6 expressed at native levels is unable to complement the mutant. Therefore, we feel that both manuscripts support each other but also bring in individual important novel aspects.

While the experiments of Gao et al are nicely designed and for example the effect of Atg1/ Atg13 and ATP on Ykt6 is directly measured in an in vitro assay, the experiments of Barz et al only focus on mutated versions of Ykt6.

We have done this experiment (n=3) and have included it in the new Appendix Figure S1F. We also refer to this experiment in the revised text:

“Also, addition of purified Atg1 complexes to the reaction abolished fusion (Appendix Figure S1F).”

However, Atg1 is known to regulate several proteins in autophagy like Atg4. It is unclear how long Atg4 is kept inactive by Atg1 to avoid premature delipidation of Atg8. It is also unclear if other factors are regulated by Atg1 during fusion. Thus, we believe that **using the D and A mutants directly in the fusion assay is a more direct way to show where and how Atg1 controls the fusion process: by regulating Ykt6**. Hence, we on purpose concentrated on the phospho-mimicking mutants of Ykt6 to show the specific and direct effect of this modification in fusion. The ATP dependence has been tested and reported earlier (Bas et al., JCB 2018; Hollenstein et al. 2019) and has been included here.

The lack of GFP-Atg8 accumulation in the Ykt6 phosphomimetic mutants argues against the conclusions drawn.

Ykt6 mutants have both a formation and a fusion defect. Therefore, in a protease protection assay in vam3D mutants one sees the autophagosome formation defect, as expected when autophagosome formation is slowed down.

This is also nicely reflected in the *in vivo* assay looking at GFP-Atg8 dot formation in wild type and vam3D fusion deficient cells: Whereas more dots are visible for the Ykt6 mutant in fusion competent cells, which indicates a fusion defect, the number of GFP dots doesn't increase as much as in a wild type cell when fusion is blocked by vam3D, suggesting that also formation of autophagosomes is reduced. This is in agreement with all our other

experiments. It is also the reason why the role in fusion cannot be studied *in vivo*, and we turned to the *in vitro* setup to clearly address the fusion step only. As we purify autophagosomes at permissive temperature in the *in vitro* setup, these autophagosomes are fully matured. We and others have shown previously, that ykt6ts mutants are fully functional in autophagy at permissive temperature (Bas et al. JCB 2018, Figure 6B).

Therefore, what we see here is absolutely expected and in agreement with all other data. We believe that Ykt6 plays a role both in autophagosome formation and during fusion. In this manuscript, however, we focus mainly on the role of Ykt6 in fusion. We have carefully explained this in the text (Page 10 of the manuscript):

"As expected, the number of GFP-Atg8 puncta per cell increased from around 2 in Vam3-containing wild type cells to about 7 in fusion-defective vam3Δ cells, indicating that in Vam3-containing cells autophagosomes are turned over by fusion with the vacuole after 1 hour of starvation (Figure 4A,B). Whereas the 3A mutant showed a similar increase in GFP-Atg8 puncta per cell, all phospho-mimicking mutants showed little increase, suggesting an autophagosome-vacuole fusion defect. Furthermore, we noticed that ykt6ts vam3Δ mutants expressing the Ykt6 phospho-mimicking variants accumulated less GFP-Atg8 compared to wild type or 3A expressing cells, which confirmed our previous findings that autophagosome formation is also affected in the Ykt6 phospho-mimicking mutants (Figure 3B). Taken together, Ykt6 phosphorylation results in both an autophagosome formation as well as a severe autophagosome-vacuole fusion defect."

In sum, Gao et al provided significant novel informations about the recruitment and regulation of Ykt6, while the manuscript by Barz et al does not meet the criteria for publication in EMBO Rep. and is solely repetitive.

We are aware of the partial overlap, as these studies were performed in parallel without knowing of each other. We disagree with the statement on lacking unique aspects. The two stories strongly support each other, but also complement each other in additional aspects. Whereas Gao et al. have addressed the role of the DSL complex, we add the **biological explanation why the phosphorylation of Ykt6 impairs autophagosome-vacuole fusion, namely by inhibiting SNARE bundling**. This has never been shown before and is the crucial finding to explain the defects observed, which would otherwise remain elusive. Also, in contrast to Gao et al. **we used functionally tagged Ykt6 expressed at endogenous levels**, whereas Gao et al. overexpressed GFP-tagged Ykt6, which is unable to complement the mutant. Therefore, we feel that both manuscripts support each other but also bring in individual important novel aspects.

The manuscript further reads as it was put together in a rush, for example the references are listed in double, but in different versions.

As appreciated by Reviewer 1, this manuscript was very carefully written and also all experiments have been well controlled. We are sorry that the references were listed in double, and have corrected this mistake.

Prof. Claudine Kraft
University of Freiburg
Institute for Biochemistry and Molecular Biology, ZBMZ
Freiburg
Germany

Dear Claudine,

Thank you for sending all revised files. I am very pleased to accept your manuscript for publication in the next available issue of EMBO reports. Thank you for your contribution to our journal.

At the end of this email I include important information about how to proceed. Please ensure that you take the time to read the information and complete and return the necessary forms to allow us to publish your manuscript as quickly as possible.

As part of the EMBO publication's Transparent Editorial Process, EMBO reports publishes online a Review Process File to accompany accepted manuscripts. As you are aware, this File will be published in conjunction with your paper and will include the referee reports, your point-by-point response and all pertinent correspondence relating to the manuscript.

If you do NOT want this File to be published, please inform the editorial office within 2 days, if you have not done so already, otherwise the File will be published by default [contact: emboreports@embo.org]. If you do opt out, the Review Process File link will point to the following statement: "No Review Process File is available with this article, as the authors have chosen not to make the review process public in this case."

Should you be planning a Press Release on your article, please get in contact with emboreports@wiley.com as early as possible, in order to coordinate publication and release dates.

Please note that under the DEAL agreement of German scientific institutions with our publisher Wiley, your paper might be eligible for open access publication in a way that is free of charge for the authors. Please contact either the administration at your institution or our publishers at Wiley (emboreports@wiley.com) for further questions.
<https://authorservices.wiley.com/author-resources/Journal-Authors/open-access/affiliation-policies-payments/institutional-funder-payments.html>

Thank you again for your contribution to EMBO reports and congratulations on a successful publication. Please consider us again in the future for your most exciting work.

Kind regards,
Martina

THINGS TO DO NOW:

You will receive proofs by e-mail approximately 2-3 weeks after all relevant files have been sent to our Production Office; you should return your corrections within 2 days of receiving the proofs.

Please inform us if there is likely to be any difficulty in reaching you at the above address at that time. Failure to meet our deadlines may result in a delay of publication, or publication without your corrections.

All further communications concerning your paper should quote reference number EMBOR-2020-51869V3 and be addressed to emboreports@wiley.com.

Should you be planning a Press Release on your article, please get in contact with emboreports@wiley.com as early as possible, in order to coordinate publication and release dates.

Corresponding Author Name: Claudine Kraft

Manuscript Number: EMBOR-2020-51869V2